



# MICS-Asia III: Overview of model inter-comparison and evaluation of acid deposition over Asia

Syuichi Itahashi[1], Baozhu Ge[2,3,4], Keiichi Sato[5], Joshua S. Fu[6], Xuemei Wang[7], Kazuyo Yamaji[8], Tatsuya Nagashima[9], Jie Li[2,3,4], Mizuo Kajino[10,11], Hong Liao[12,13], Meigen Zhang[2,3,4], Zhe Wang[14], Meng Li[15],
Junichi Kurokawa[5], Gregory R. Carmichael[16], Zifa Wang[2,3,4]

[1] Environmental Science Research Laboratory, Central Research Institute of Electric Power Industry (CRIEPI), Abiko, Chiba 270–1194, Japan
[2] State Key Laboratory of Atmospheric Boundary Layer Physics and Atmospheric Chemistry (LAPC), Institute of Atmospheric Physics (IAP), Chinese Academy of Sciences (CAS), Beijing 100029, China
[3] Collage of Earth Science, University of Chinese Academy of Sciences, Beijing 100049, China
[4] Center for Excellence in Urban Atmospheric Environment, Institute of Urban Environment, Chinese Academy of Sciences (CAS), Xiamen 361021, China
[5] Asia Center for Air Pollution Research (ACAP), 1182 Sowa, Nishi-ku, Niigata, Niigata 950–2144, Japan
[6] Department of Civil and Environmental Engineering, University of Tennessee, Knoxville, TN 37996, USA
[7] Institute for Environment and Climate Research, Jinan University, Guangzhou 510275, China
[8] Graduate School of Maritime Sciences, Kobe University, Kobe, Hyogo 658–0022, Japan
[9] National Institute for Environmental Studies (NIES), Tsukuba, Ibaraki 305–8506, Japan
[10] Meteorological Research Institute (MRI), Tsukuba, Ibaraki 305–0052, Japan
[11] Faculty of Life and Environmental Sciences, University of Tsukuba, Tsukuba, Ibaraki 305–8572, Japan
[12] School of Environmental Science and Engineering, Nanjing University of Information Science & Technology, Nanjing 210044, China
[13] International Joint Laboratory on Climate and Environmental Change, Nanjing University of Information Science & Technology, Nanjing 210044, China
[14] Research Institute for Applied Mechanics (RIAM), Kyushu University, Kasuga, Fukuoka 816–8580, Japan
[15] Ministry of Education Key Laboratory for Earth System Modeling, Department of Earth System Science, Tsinghua University, Beijing 100084, China
[16] Center for Global and Regional Environmental Research, University of Iowa, Iowa City, IA 52242, USA

*Correspondence to*: Syuichi Itahashi (isyuichi@criepi.denken.or.jp)

**Abstract.** The Model Inter-Comparison Study for Asia (MICS-Asia) Phase III was conducted to promote understanding of regional air quality and climate change in Asia, which have received growing attention due to the huge amount of anthropogenic emissions worldwide. This study provides an overview of acid depositions. Specifically, dry and wet depositions of the following species were analyzed: S (sulfate aerosol, sulfur dioxide ($SO_2$), and sulfuric acid ($H_2SO_4$)), N (nitrate aerosol, nitrogen monoxide (NO), nitrogen dioxide ($NO_2$), and nitric acid ($HNO_3$)), and A (ammonium aerosol and ammonia ($NH_3$)).
The wet deposition simulated by a total of nine models was analyzed and evaluated using ground observation data from the Acid Deposition Monitoring Network in East Asia (EANET). In this Phase III study, the number of observation sites was increased to 54 from 37 in the Phase II study, and Southeast Asian countries were newly added. Additionally, whereas the analysis period was limited to representative months of each season in MICS-Asia Phase II, this Phase III study analyzed the



full year of 2010. The scope of this overview mainly focuses on the annual accumulated depositions. In general, models can capture the observed wet depositions over Asia but underestimate the wet deposition of S and A and show large differences in the wet deposition of N. Furthermore, the ratio of wet deposition to the total deposition (the sum of dry and wet deposition) was investigated in order to understand the role of important processes in the total deposition. The general dominance of wet

deposition over Asia and attributions from dry deposition over land were consistently found in all models. Then, total deposition maps over 13 countries participating in EANET were produced, and the balance between deposition and anthropogenic emissions was calculated. Excesses of deposition, rather than of anthropogenic emissions, were found over Japan, North Asia, and Southeast Asia, indicating the possibility of long-range transport within and outside Asia, as well as other emission sources. To improve the ability of models to capture the observed wet deposition, two approaches were

attempted, namely, ensemble and precipitation adjustment. The ensemble approach was effective at modulating the differences in performance among models, and the precipitation-adjusted approach demonstrated that the model performance for precipitation played a key role in better simulating wet deposition. Finally, the lessons learned from this Phase III study and future perspectives for Phase IV are summarized.

## 1 Introduction

With recent increases in anthropogenic emissions, Asia has experienced the highest atmospheric acid deposition worldwide (Vet et al., 2014). Atmospheric concentrations and depositions are monitored in the USA by the Clean Air Status and Trends Network(CASTNET, 2019) and in Europe by the European Monitoring and Evaluation Programme (EMEP, 2019). Over Asia, the Acid Deposition Monitoring Network in East Asia (EANET, 2019a) has maintained a regular observation network since 2000 to measure and understand acid deposition in Asia. The detailed findings from EANET have been reported in its first

(EANET, 2006a,b), second (EANET, 2011a,b), and third (EANET, 2016a,b) periodic reports and in a scientific review (EANET, 2015). Data are also available to the public (EANET, 2019b).

Atmospheric pollutants and depositions have been shown to be affected not only by local sources but also by long-range transport. Observation is of course important to measure phenomena in the atmosphere at specific sites. However, there is sometimes difficulty in interpreting such phenomena due to the complex impacts of both nearby and distant sources. Chemical

transport models (CTMs) representing the fate of atmospheric pollutants from emissions, transport, chemical reactions, and depositions have been recognized as valuable tools for modern atmospheric environmental sciences. Although CTMs are based on state-of-the-art science, their uncertainties should be considered (Carmichael et al., 2008a). An interpretation based on one CTM can cause misunderstanding of phenomena due to its uncertainty. To further our understanding of CTMs over Asia, the Model Inter-Comparison Study for Asia (MICS-Asia) Phase I was conducted in 1998–2000 (Carmichael et al., 2002) and

MICS-Asia Phase II was conducted in 2003–2008 (Carmichael et al., 2008b). Phase I focused on sulfur concentrations and deposition due to long-range transport in January and May 1993. A total of eight Eulerian and Lagrangian models were used.



Observation datasets of sulfur dioxide ($SO_2$) and sulfate aerosol concentrations and wet sulfate aerosol ($SO_4^{2-}$) deposition were prepared by a cooperative monitoring network in East Asia (Fujita et al., 2000), and a total of 18 sites located in China, Taiwan, Republic of Korea, and Japan were compared with models. Estimates of deposition were consistent among different models, but varied by a factor of 5 at some locations. The reason for this variability was determined to be emissions and the underlying meteorological field. It was also found that the model structure of vertical resolution was more important than the parameterization used in the chemical conversion and removal processes. MICS-Asia Phase II focused on oxidants (Ox) and particulate matter (PM) (Carmichael et al., 2008b). In terms of depositions, the dry deposition of $SO_4^{2-}$, nitrate ($NO_3^-$), and ammonium ($NH_4^+$) aerosol and the relevant gas species of $SO_2$, nitric acid ($HNO_3$), and ammonia ($NH_3$), and the wet deposition of $SO_4^{2-}$, $NO_3^-$, and $NH_4^+$, were compared among eight Eulerian models. To compare the seasonality of depositions, the four periods of March, July, and December 2001 and March 2002 were analyzed. The EANET observation data of wet deposition at 37 sites were compared with models (Wang et al., 2008). The models generally reproduced acid depositions in China, Korea, Japan, and Southeast Asia, but could not accurately describe depositions in inland areas such as Mongolia and Russia. These differences were attributed to differences in meteorology, chemical mechanisms, and deposition parameterizations. In Phase II, the ensemble-mean depositions over East Asia based on eight models were determined for the first time, and showed better skill than any single model. Additionally, in Phase II, emission data were made to be uniform to remove potential discrepancies, but participant models used different modeling domains with different horizontal and vertical structures and different meteorological models. Here, we present MICS-Asia Phase III. This phase consists of three parts: Topic 1, the comparison and evaluation of current multi-scale air quality models; Topic 2, the development of reliable emission inventories in Asia; and Topic 3, the interactions between air quality and climate change. Scientific papers have been published focusing on gas-phase species (Kong et al., 2019), aerosols (Chen et al., 2019), and ozone ($O_3$) (Akimoto et al., 2019; Li et al., 2019). The details of Topic 2 (Li et al., 2017) and Topic 3 (Gao et al., 2018) have also been reported.

This paper is concerned with Topic 1 and focuses on depositions—namely, the output process from the atmosphere together with the input process to the surface. This manuscript is structured as follows. Section 2 introduces the framework of the model inter-comparison study for deposition. Models and observations are respectively described in Sections 2.1 and 2.2. Section 3 is dedicated to results. Section 3.1 presents an evaluation of precipitation and Section 3.2 presents an evaluation of wet deposition. Section 4 provides an in-depth discussion of the analysis results. First, Section 4.1 presents total deposition maps over Asia. The proportion of wet deposition to total deposition over Asia was analyzed in order to clarify the relative importance of dry and wet deposition processes. By comparing the amount of anthropogenic emissions that are input to the atmosphere, the implications of other emission sources and long-range transport are discussed. Next, in Section 4.2, an ensemble approach is applied to combine multi-model results. Then, in Section 4.3, a precipitation-adjusted approach is applied with the aim of further improving model performance for simulating wet deposition. Finally, Section 5 summarizes this research and puts forward future perspectives for MICS-Asia Phase IV.



## 2 Framework of model inter-comparison for deposition

### 2.1 Model description

In MICS-Asia Phase III, all participating models were run for the year 2010 and requested to submit simulations of the monthly accumulated amounts of dry and wet depositions of S species (sulfate aerosol, $SO_2$, $H_2SO_4$), N species (nitrate aerosol, NO,

$NO_2$, $HNO_3$), and A species (ammonium aerosol and $NH_3$). A total of nine models (M1, M2, M4, M5, M6, M11, M12, M13, and M14; these numbers are unified in MICS-Asia Phase III activities) were used in this deposition analysis. These models and their configurations are summarized in Table 1. This study used four different CTMs: the Community Multiscale Air Quality (CMAQ) modeling system (Byun and Schere, 2006), developed by the US Environmental Protection Agency (EPA); the nested air quality prediction model system (NAQPMS), developed by the Institute of Atmospheric Physics (IAP) of the

Chinese Academy of Sciences (CAS) (Wang et al., 2001; Li et al., 2016; Ge et al., 2016); the non-hydrostatic mesoscale model coupled with chemistry transport model (NHM-Chem), developed by the Meteorological Research Institute (MRI) in Japan (Kajino et al., 2018, 2019); and the global three-dimensional model of atmospheric chemistry driven by meteorological input from the Goddard Earth Observing System with chemistry (GEOS-Chem), developed by Harvard University (Bey et al., 2001). Basically, Phase III was conducted with a unified domain and meteorological field as a "standard" setting based on experience

in Phases I and II. The modeling domain covers the whole of Asia with a horizontal grid resolution of 45 km by 182 × 172 grids on a Lambert conformal projection, as shown in Fig. 1, and with 40 vertical layers from the surface to 10 hPa. The meteorological fields were driven by the Weather Research and Forecasting (WRF) model version 3.4.1 (Skamarock et al., 2008). Analysis nudging was conducted using the National Centers for Environmental Prediction and National Center for Atmospheric Research (NCEP/NCAR) final analysis data (FNL) (http://rda.ucar.edu/datasets/ds083.2/), available with a 1° ×

1° horizontal resolution and 6 h temporal resolution for temperature, wind, and water vapor. For this deposition analysis, seven of the nine models (M1, M2, M4, M5, M6, M11, and M12) were configured with the same domain and meteorological conditions. One model (M13) was configured with a horizontal grid resolution of 0.5° × 0.667° covering part of Asia (70−150° E, 11° S−55° N) with 47 vertical layers from the surface to 0.01 hPa. This model's meteorological fields were driven by the assimilated meteorological fields from the Goddard Earth Observing System 5 (GEOS5) of the US National Aeronautics and

Space Administration (NASA) (https://gmao.gsfc.nasa.gov). Another model (M14) covered a smaller domain compared with the "standard", with a horizontal grid resolution of 45 km and 15 vertical layers. The meteorological fields for this model were simulated by the Regional Atmospheric Modeling System (RAMS) (Pielke et al., 1992) using FNL for analysis nudging as in the "standard" WRF simulation. As input data, emissions were unified for all models. The anthropogenic emissions over Asia were taken from the MIX anthropogenic emission inventory developed for MICS-Asia Phase III (Li et al., 2017). This

inventory was developed by harmonizing emission inventories over Asia using a mosaic approach. The base inventory was the Regional Emission Inventory in Asia (REAS) version 2.1 (Kurokawa et al., 2013), with replacement over China using the Multi-Resolution Emission Inventory for China (MEIC), developed by Tsinghua University, and a high-resolution $NH_3$ inventory developed by Peking University (Huang et al., 2012); replacement over Korea using the Clean Air Policy Support



System (CAPSS) (Lee et al., 2011); replacement over Japan using the Japan Auto-Oil Program (JATOP) (JPEC, 2012a,b,c); and replacement over India using an Indian inventory developed by Argonne National Laboratory (Lu et al., 2011; Lu and Streets, 2012). Based on the WRF meteorological field, hourly biogenic emissions were calculated by the Model of Emissions of Gases and Aerosols from Nature (MEGAN) version 2.04 (Guenther et al., 2006). Emissions from biomass burning were

taken from the Global Fire Emission Database (GFED) version 3 (van der Werf et al., 2010). $SO_2$ emissions from volcanoes were obtained from the AEROCOM program (AEROCOM, 2019). These emissions were unified and provided as an input dataset, and the temporal variation and vertical allocation were requested to follow the setup table; however, there seems to have been a mismatch during their process for each model. For example, the differences in NO emissions intensity were reported in our companion paper focusing on $O_3$ analysis (Li et al., 2019). In this sense, this Phase III was conducted not as a

model inter-comparison but rather as a modeling system inter-comparison. This is one of the lessons learned from this Phase III study, and is one of our future research subjects to provide a single model-ready emission file; however, this can propose the potential model variation caused by the setup of the modeling system.

The models used for this deposition analysis were configured with various physical (advection and diffusion scheme) and chemical (gas and aerosol chemistry) processes. The physical model setup for horizontal and vertical advection and diffusion

processes was based on their CTM, and CMAQ provides some options to choose them; hence, setups are different even in the same model version. In the chemical scheme, gas-phase chemistry was configured by SAPRC-99 (Carter et al., 2000) to treat 76 species with 214 reactions for M1, M2, M4, M5, M6, M12, and M14, CBMZ (Zaveri and Peters, 1999) including 67 species and 164 reactions for M11, and the developed scheme in GEOS-Chem (Bey et al., 2001) constructed with 80 species and 300 chemical reactions for M13. Aerosol treatment in the models was AERO5 or AERO6 in the CMAQ models of M1, M2, M4,

M5, M6, and M14, and the representations of secondary organic aerosol (SOA) in these are respectively documented in their evaluation documents for AERO5 (Carlton et al., 2010) and AERO6 (Simon and Bhave, 2012). A bulk yield scheme to treat six SOAs has been embedded in M11 (Li et al., 2011). The unique options for aerosol representations with five-category non-equilibrium, three-category non-equilibrium, and bulk equilibrium for research on climate, air pollution, and operational forecasts were available in M12, and details have been reported elsewhere (Kajino et al., 2018, 2019). The originally developed

scheme for M13 in GEOS-Chem is available in the literature (Park, 2004; Pye et al., 2009). Regarding the thermodynamic equilibrium to treat inorganic aerosol species, the ISORROPIA model was used for all models but with different versions, namely, version 1.7 (Nenes et al., 1998) and the updated version 2.1 to further treat trace metals (Fountoukis and Nenes, 2007). CTMs configured the dry deposition process based on the resistance-in-series model (Wesely and Hicks, 1977). Models M11 and M13 were based on the traditional scheme of Wesely (1989), with numerous modifications for M13 (Wang et al., 2004),

and CMAQ models of M1, M2, M4, M5, M6, and M14 used the M3DRY scheme (Pleim et al., 2001). The dry deposition scheme in M12 was extensively modified to include the updated observation data; details can be found in the description paper (Kajino et al., 2018, 2019). For the wet deposition process, theoretically, all models use Henry's law to entrain the air pollutants into cloud. The sequential wet deposition process and related aqueous phase chemistry were based on the Regional Acid



Deposition Model (RADM) (Chang et al., 1987). In the case of CMAQ, there is an improvement in the treatment of the precipitation flux before and after the release of version 4.7 (Foley et al., 2010). CMAQ uses an algorithm to allocate precipitation amounts to individual layers based on a normalized profile of hydrometeorological components of rain, snow, and graupel. Before version 4.7, CMAQ allocated the precipitation flux into vertical layers without taking into account the

layer thickness; hence, many air pollutants were removed from thin layers but fewer air pollutants were removed from thick layers. In version 4.7, this point was revised to compute the precipitation flux for each layer as a function of the non-convective precipitation, the sum of hydrometers, and the layer thickness. This difference might be found in M14 and other CMAQ models; however, the input meteorological data are different for RAMS and WRF, and it is difficult to detect the effect of this difference. A similar approach with CMAQ is also taken in M11. The details of wet deposition processes are described in the

description papers for model M12 (Kajino et al., 2018, 2019). The wet deposition scheme in M13 has been tested through the combined use of terrigenic $^{210}$Pb and cosmogenic $^{7}$Be (Liu et al., 2001). MICS-Asia Phase III provides two sets of lateral boundary conditions derived from the three-hourly global model outputs of GEOS-Chem (Bey et al., 2001) and CHASER (Sudo et al., 2002a,b); GEOS-Chem was run with 2.5º × 2º resolution and 47 vertical layers and CHASER was run with 2.8º × 2.8º resolution and 32 vertical layers, and the participants can choose between them. Models M1, M13, and M14 used GEOS-

Chem, and models M4, M5, M6, M11, and M12 used CHASER. Model M2 used the default boundary condition field provided in the CMAQ modeling system.

## 2.2 EANET observations

In this overview paper, model evaluations are based on EANET observations over Asia. In EANET, wet deposition is measured by a wet-only sampler designed to collect precipitation samples during rainfall. The locations of observation sites are shown

in Fig. 1. Table 2 lists detailed information including latitude (°N), longitude (°E), altitude (meters above sea level (a.s.l.)), and the classification, sampling interval, and analysis method for anions and cations at each site. In EANET, site classification is defined as follows: urban sites are defined as urbanized and industrial areas or the areas immediately outside them; rural areas are defined as those more than 20 km away from large pollution sources; and remote areas are defined as those more than 50 km away from large pollution sources and more than 500 m away from main roads. The sampling intervals were

different from site to site, and a monthly accumulated dataset was used for the model evaluation. The analysis method for anions ($SO_4^{2-}$ and $NO_3^-$) and cations ($NH_4^+$) was based on ion chromatography at most sites, with some exceptions in Russia. The observed data were checked by ion balance and conductivity agreement, and the completeness of the data was determined from the duration of precipitation coverage and total precipitation amount. More details can be found in EANET manuals (EANET, 2000, 2010a).

To compare and evaluate the model simulation results with EANET observations, we used the statistical metrics of correlation coefficient (R), normalized mean bias (NMB), and normalized mean error (NME), which are defined as follows:


$$R = \frac{\sum_1^N |(O_i - \bar{O})(M_i - \bar{M})|}{\sqrt{\sum_1^N (O_i - \bar{O})^2}\sqrt{\sum_1^N (M_i - \bar{M})^2}} \qquad (1)$$

$$NMB = \frac{\sum_1^N (M_i - O_i)}{\sum_1^N O_i} \qquad (2)$$

$$NME = \frac{\sum_1^N |M_i - O_i|}{\sum_1^N O_i} \qquad (3)$$

Here, N is the total number of paired observations (O) and models (M). Furthermore, in order to judge the agreement between

simulation and observation, the percentages within a factor of 2 (FAC2) and within a factor of 3 (FAC3) were also calculated.

For the analysis of the total amount of deposition over Asia, 13 countries in Asia participating in EANET activities were targeted, as shown in Fig. 1. These 13 countries were divided into four regions: East Asia (China, Republic of Korea, and Japan); North Asia (Mongolia and Russia), continental Southeast Asia (Myanmar, Thailand, Lao People's Democratic Republic (PDR), Cambodia, and Vietnam); and oceanic Southeast Asia (the Philippines, Malaysia, and Indonesia).

**3 Results**

**3.1 Evaluation of precipitation**

In MICS-Asia Phase III, the meteorological field was simulated by WRF for models M1, M2, M4, M5, M6, M11, and M12, by GEOS5 for model M13, and by RAMS for model M14. Before the evaluation of wet deposition, here we evaluated the precipitation amount based on the monthly accumulated value. Precipitation data were also taken from EANET observation

sites. Figure 2 shows scatter plots of monthly accumulated precipitation amount between observation and models (WRF, GEOS5, and RAMS), with different symbols for different countries and different colors for different months. For WRF, modeled precipitation slightly overestimated the observed precipitation amount, NMB was +12.3%, and NME was +64.2% with an R of 0.62. An overestimation of precipitation by more than a factor of 3 can be found for Southeast Asian countries in wintertime, and an underestimation of precipitation by more than a factor of 3 can be found in East Asia. For GEOS5, the

agreement with observation was better than for WRF; in particular, there was less model underestimation with GEOS5, but the overestimation for Southeast Asian countries in wintertime was similar for both GEOS5 and WRF. The statistical performance was as follows: R of 0.71, NMB of +3.2%, and NME of +46.5%. For RAMS, due to its smaller domain compared with WRF, the observation number was small; it did not contain data over Malaysia or Indonesia. In contrast to WRF and GEOS5, RAMS showed a general underestimation, with NMB of -8.8%. The R and NME values of RAMS were comparable

with those of WRF. In summary, the monthly accumulated precipitation amounts were captured overall by three different meteorological models. However, it should be noted that overestimation (underestimation) could lead to the overestimation (underestimation) of wet deposition. In relation to this point, we applied a precipitation-adjusted approach within the framework of MICS-Asia Phase III; this is discussed in Section 4.3.



### 3.2 Evaluation of wet deposition

#### 3.2.1 Wet deposition of S

A comparison of model performance for each of the 54 EANET observation sites is shown in Fig. 3 (a), and a statistical analysis based on monthly accumulated wet deposition is summarized in Table 3. The available numbers for this comparison are smaller compared with the precipitation data (Fig. 2 inset) due to the small number of observations after the quality checking and assurance process. Compared with EANET observation over Asia, it was found that the models tended to underestimate the wet deposition of S, except model M11. Regarding statistical scores, values of R were around 0.4, values of NMB were around -30%, except for model M11, and values of NME were around 70%, except for model M11. For model M11, the value of NMB was +44.3% and that of NME was +136.1%, showing the overestimation of wet deposition of S, in contrast to other models. The performances of the CMAQ models M1, M2, M4, M5, and M6 were similar. The R scores were higher in models M13 and M14 than in other models; these two models were driven by different meteorological models. The underestimation of wet deposition by model M14 was greater than that by the other models, which partly stemmed from the underestimation of precipitation (Section 3.1.1). Approximately 40% of the model simulation results were within FAC2, and 60% were within FAC3. For model M11, about 30% and 40% of simulation results were within FAC2 and FAC3, respectively. The model performances for atmospheric concentrations were presented in our companion paper (Chen et al., 2019). Generally, the models used in this deposition analysis underestimated atmospheric concentrations of $SO_4^{2-}$ over Asia; however, M14 was distinguished by the overestimation of the atmospheric concentration of $SO_4^{2-}$ over coastal regions, such as over Korea and Japan (Fig. 3 of Chen et al., 2019). This overestimation is related to the underestimation of $SO_4^{2-}$ wet deposition that is also found in M14 (Fig. 3). The seasonal pattern of atmospheric $SO_4^{2-}$ concentration showed underestimation, especially in winter (Fig. 5 of Chen et al., 2019). This was also revealed by a model inter-comparison study in Japan, and suggests the insufficient aqueous-phase production of $SO_4^{2-}$, especially in winter (Itahashi et al., 2018b,c). Moreover, another study pointed out the importance of heterogeneous chemistry as a possible explanation for the missing production of $SO_4^{2-}$ (Shao et al., 2019).

The spatial distribution of the wet deposition of S is displayed in Fig. 4. Model results showed the highest amount of wet deposition (dark red, over 20 kg S ha$^{-1}$ year$^{-1}$) over Eastern China and the island of Java, Indonesia, and moderately high amounts (yellow to orange, 10–20 kg S ha$^{-1}$ year$^{-1}$) over the Korean Peninsula, Japan and surrounding oceans, and India. A comparison of the model performances for 13 countries is shown in Fig. 3 (b). Based on EANET observations, China had the highest amount of wet deposition, with 21.9 kg S ha$^{-1}$ year$^{-1}$, followed by Indonesia, with 12.4 kg S ha$^{-1}$ year$^{-1}$, Vietnam, with 11.4 kg S ha$^{-1}$ year$^{-1}$, and Japan, with 10.6 kg S ha$^{-1}$ year$^{-1}$. Other countries had amounts of wet S deposition below 10 kg S ha$^{-1}$ year$^{-1}$. Compared with EANET observation data, all the models except M11 underestimated wet deposition in China, whereas M11 overestimated it. The model performances for China were similar to those for Korea and Japan. For Indonesia, wet deposition was underestimated by model M11 and overestimated by M12, while the estimations of the other models were comparable to the observed values. Conversely, model M11 overestimated wet deposition in East Asia, but performed similarly



in Vietnam and other Southeast Asian countries as it did for Indonesia. As summarized in Fig. 3 (c), model M11 overestimated wet deposition over East and North Asia, and underestimated it over Southeast Asia. Other models exhibited underestimation over East and North Asia and continental Southeast Asia. The wet deposition over oceanic Southeast Asia was well captured by all models except M11 and M12, with M12 showing overestimation. The model performance divided into three EANET

classifications is also shown in Fig. 3 (d). The observation results showed a wet S deposition of 11.8, 11.2, and 7.8 kg S ha$^{-1}$ year$^{-1}$ at urban, rural, and remote sites, respectively. This result demonstrates the similarly polluted wet deposition status of urban and rural EANET sites. All models showed a strong decrease in wet deposition at urban sites compared with rural sites, but a weak decrease in wet deposition between remote and rural sites. However, since the definition of rural areas is those more than 20 km away from large pollution sources, the horizontal resolution of 45 km has difficulty fully capturing these

spatial scales; therefore, a finer-scale simulation will be of interest in future study.

### 3.2.2 Wet deposition of N

The model performances for the wet deposition of N were compared with EANET observations; the results for each EANET observation site are shown in Fig. 5 (a). The model performance for the wet deposition of N exhibited larger variation among models compared with the wet deposition of S. Even in the same model of CMAQ, models M1, M2, M4, M5, and M6 showed

different performances for the wet deposition of N. The model performances for the atmospheric concentration of NO$_3^-$ in our companion paper also showed large differences between models (Fig. 3 of Chen et al., 2019). Over Asia, models M2, M4, M5, and M6 showed underestimation, whereas models M1, M11, M12, M13, and M14 showed overestimation (Table 2 of Chen et al., 2019). Over the coastal regions of Korea and Japan, models strongly overestimated the atmospheric concentration of NO$_3^-$ (Fig. 5 of Chen et al., 2019). If both H$_2$SO$_4$ and HNO$_3$ are present, H$_2$SO$_4$ preferentially reacts with NH$_3$, and therefore NH$_4$NO$_3$

is produced only if excess NH$_3$ is present. The underestimation of the atmospheric concentration of SO$_4^{2-}$ can lead to the overestimation of the atmospheric concentration of NO$_3^-$. Over China, models underestimated the production of both SO$_4^{2-}$ and NO$_3^-$ (Fig. 5 of Chen et al., 2019). This may be partly related to the general underestimation of NH$_3$ concentration over China (Table 2 and Figs. 2 and 3 of Kong et al., 2019).

The spatial distribution of the wet deposition of N is shown in Fig. 6. With respect to reactive nitrogen (Nr) deposition, a

threshold value of 10 kg N ha$^{-1}$ has been established (e.g., Bleeker et al., 2011). The exceeding of this threshold value (from yellow to orange and red) was different among models. Models M1, M5, and M11 simulated excess Nr deposition over Eastern China and the Sea of Japan, while models M4, M6, M12, and M13 simulated more limited areas of excess Nr deposition and models M2 and M14 showed no excess Nr deposition over Asia. A statistical analysis of model performances for the wet deposition of N is presented in Table 4. Compared with EANET observations, models M5 and M11 overestimated the wet

deposition of N, with NMB values of +16.5% and +51.9%, respectively; model M1 simulated values that are almost comparable, with NMB of -3.4%; and other models underestimated the wet deposition of N, with negative NMB values; models M4, M6, M12, and M13 showed NMB values of around -30%, and models M2 and M14 showed NMB values below



-50%. NME values were greater than +50% for all models. R values ranged from 0.32 to 0.42 for all models driven by WRF, while models M13 and M14, which were simulated with a different meteorological field, showed higher R values of 0.62 and 0.51, respectively. This feature of higher R values for models M13 and M14 is similar to the finding for the wet deposition of S. Approximately 40% of the model simulation results were within FAC2, and 60% were within FAC3. Model M13 showed

better agreement, with 54.3% and 70.6% of its simulation results being within FAC2 and FAC3, respectively. A comparison between country and regional summarized results and EANET observations is shown in Fig. 5 (b) and (c). These showed large variability. Over East Asia, the observed wet depositions of N were 6.8, 6.7, and 3.3 kg N ha$^{-1}$ year$^{-1}$ for China, Korea, and Japan, respectively, with an average value of 4.9 kg N ha$^{-1}$ year$^{-1}$. It was also found that models M11, M5, and M1 overestimated, and models M14 and M2 underestimated, wet deposition. A similar pattern was also observed in North Asia.

These results suggest difficulty in the exact estimation of the wet deposition of N, and hence in the setting of the threshold value of Nr. Unlike the situation for East and North Asia, all of the models underestimated the wet deposition of N over both continental and oceanic Southeast Asian countries. A detailed analysis of site classification was also performed for wet N deposition, as shown in Fig. 5 (d). Observations indicate a wet deposition of N of 5.2, 4.8, and 2.4 kg N ha$^{-1}$ year$^{-1}$ at urban, rural, and remote sites, respectively, similar to the findings for the wet deposition of S. Although all models showed

underestimation for urban and rural sites, comparable levels of wet deposition over urban and rural sites were simulated by all models. However, the modeled wet N depositions were relatively similar for remote sites, and did not reproduce the observed decrease in the wet deposition of N at remote sites compared with rural sites. The exception to this was model M13, which showed a decrease in the wet deposition of N at remote sites compared with rural sites.

### 3.2.3 Wet deposition of A

Wet deposition of A was compared between model simulations and EANET observations, and the results for observation sites are shown in Fig. 7 (a). The statistical analyses for the wet deposition of A are listed in Table 5. As $NH_4^+$ is the counterpart of $SO_4^{2-}$ and $NO_3^-$, the behavior of the wet deposition of A showed a blend of features of the wet depositions of S and N. In general, model simulations underestimated the wet deposition of A, except model M13. NMB values ranged from -49.3% for model M14 to -0.2% for model M12 to +8.6% for model M13. NME values were around +70%. R values were around 0.3,

while models M13 and M14, which were simulated with a different meteorological field, showed higher R values (0.48 and 0.54, respectively), as was also observed for the wet depositions of S and N. Approximately 40% of the model simulation results were within FAC2, and less than 60% were within FAC3. Model M13 showed better agreement, with 50.9% and 68.3% of simulation results being within FAC2 and FAC3, respectively. Our companion paper reported model performances for the atmospheric concentration of $NH_4^+$ as well as model performances associated with the atmospheric concentrations of $SO_4^{2-}$

and $NO_3^-$ as counter ions (Fig. 3 of Chen et al., 2019).

The simulated spatial distribution of the wet deposition of A is displayed in Fig. 8. Compared with the wet depositions of S (Fig. 4) and N (Fig. 6), the wet deposition of A above 2 kg N ha$^{-1}$ year$^{-1}$ (light blue) was more limited over land. A large





amount of wet A deposition (dark red, over 20 kg N ha$^{-1}$ year$^{-1}$) was found in areas over the central parts of China and India; these areas were simulated to be broader by model M13. The threshold value of Nr deposition of 10 kg N ha$^{-1}$ was found to be exceeded over Central China with limited spatial coverage, with model M13 simulating a broader area. Models M12 and M13 simulated a larger amount of wet A deposition over India, and also simulated an extended wet deposition of A from India to the Indian Ocean; however, we cannot judge these simulations due to the lack of observation in this region. The summarized results for countries and regions are shown in Fig. 7 (b) and (c). The underestimation of the wet deposition of A over North and East Asia and its overestimation over Southeast Asia were consistently observed in models. In particular, all models overestimated the wet deposition of A over the Philippines and Malaysia. These common features across models suggest a shortcoming in the current status model, and thus suggest the need to revisit emission inventories. As shown in Fig. 7 (d), observations showed wet deposition of A of 6.5, 6.3, and 2.4 kg N ha$^{-1}$ year$^{-1}$ at urban, rural, and remote sites, that is, similar to the observations of the wet deposition of S and N. All models showed a decrease in the wet deposition of A from urban to rural sites and from urban to remote sites, with the decreases being much greater than the observed ones. Despite the underestimation at rural sites, all models overestimated the wet deposition of A at remote sites.

## 4 Discussion

### 4.1 Total deposition mapping over Asia

Here, we draw maps of total deposition (defined as the sum of dry and wet deposition) over Asia to investigate the annual accumulated total deposition across various countries and its relation to emission amounts input to the atmosphere. In Figs. 9 to 11, the total depositions of S, N, and A over 13 countries participating in EANET are respectively mapped. In each figure, the ensemble mean, which is simply the average of all nine models, is shown in order to demonstrate the spatial distribution pattern. A detailed discussion of the ensemble approach is given in the following Section 4.2. An evaluation of the atmospheric concentrations of gases and aerosols simulated by models is given in our companion papers (Kong et al., 2019; Chen et al., 2019). Due to the difficulty of directly measuring deposition, especially within the framework of observation networks, we relied on model-simulated atmospheric concentrations to estimate amounts of dry deposition. In the supplemental material, the annual accumulated dry and total depositions of S, N, and A are respectively shown in Figs. S1 and S2, Figs. S4 and S5, and Figs. S7 and S8. The components of dry deposition averaged over domain on annual mean are listed in Tables S1 to S3 in the supplemental material. In models, $SO_2$, $HNO_3$, and $NH_3$ were commonly the most dominant species for dry deposition. To investigate the important processes in S, N, and A deposition, the proportion of wet deposition to total deposition is also illustrated in Fig. S3, S6, and S9, with blue indicating the higher proportion of wet deposition to total deposition and brown indicating the higher proportion of dry deposition to total deposition.

In terms of the total deposition of S (Fig. 9), high amounts were seen over East Asian countries: in Eastern China to the Korean Peninsula; in Japan and surrounding oceans; in Java, Indonesia; and in Eastern India. The total amount of S deposition across



China was around 10,000 Gg S year$^{-1}$ (5600–15,500 Gg S year$^{-1}$), nearly 10 times more than in the country with the second largest deposition, namely Indonesia, where deposition was around 1300 Gg S year$^{-1}$ (640–2500 Gg S year$^{-1}$). Other countries had total depositions below 1000 Gg S year$^{-1}$. The proportions of dry and wet depositions of S are shown in Fig. 9, and the proportion of the wet deposition to the total deposition of S is shown in Fig. S3 in the supplemental material. Generally, wet

deposition was the most important process in the total deposition of S. The proportion of dry deposition was found to be larger over Northern Asia, and dry deposition was found to be dominant around the Bohai Sea, the Sichuan Basin, and the northwestern boundary of the modeling domain. The proportion of wet deposition to total deposition was higher than 70% over the ocean. Models M11 and M14 simulated a greater importance of dry deposition, while model M12 simulated a greater importance of wet deposition. Regarding the balance between anthropogenic emissions and deposition amounts, all countries

in Asia except China and Korea were found to experience a deposition amount that was greater than their own anthropogenic emissions, though with some variability among models. This suggests the existence of other important sources of emissions, such as volcanic emissions in the case of SO$_2$, or the possibility of long-range transport from other countries. It should also be noted that the uncertainty in anthropogenic SO$_2$ emissions over Asia has been reported to be about ±30% (Kurokawa et al., 2013). For example, for SO$_2$, all models estimated more deposition than anthropogenic emissions in Japan. Previous studies

based on source−receiver relationships in model experiments indicated the importance of volcanic sources and trans-boundary air pollution for S over Japan (Kajino et al., 2011; Kuribayashi et al., 2012; Itahashi et al., 2017; Itahashi, 2018). Over North Asia (Mongolia and Russia), all models simulated more deposition than emissions; deposition was predicted to be 1.9 and 2.5 times higher than emissions over Mongolia and Russia, respectively, by model M2, which estimated the lowest total deposition. In North Asia, the long-range transport within the Asian domain and the effect from the northern boundary of modeling domain

(i.e., global-scale impacts) are important factors leading to the excess of depositions. Over Southeast Asia, although more deposition than emissions was predicted, there was variability between models. Over Myanmar and Cambodia, the estimated anthropogenic SO$_2$ emissions were low, at 34 and 13 Gg S year$^{-1}$, respectively, and all models estimated greater deposition than emissions, suggesting long-range transport from other countries. Over Lao PDR, only model M2 showed comparable values of deposition and emissions, while other models showed an excess of depositions. In Thailand, Vietnam, the Philippines,

Malaysia, and Indonesia, some models showed an excess of depositions compared with emissions, while others did not; these results found over Southeast Asia indicate the possible importance of long-range transport or other emission sources (e.g., volcanoes and biomass burning). However, this should be carefully interpreted, since the variability of model performances reveals the risk of policymaking relying on the result of only one model.

The map of the total deposition of N (Fig. 10) illustrates a different feature to the map of the total deposition of S (Fig. 9). The

total deposition of N is largely limited over land; this is due to the fact that high amounts of dry N deposition are limited to land (Fig. S4). A drastic reduction of the total deposition of N, from over 20 Gg N ha$^{-1}$ year$^{-1}$ (dark red) to below 10 Gg N ha$^{-1}$ year$^{-1}$ (green), is found over the eastern coastline of China. The total amount of N deposition over China ranged from 2800–7200 Gg N year$^{-1}$, with large variability among models. In the other 12 countries, total N deposition was below 1000 Gg N



year$^{-1}$, thus illuminating the serious N burden over China (Xu et al., 2015). The proportion of wet N deposition to total N deposition is shown in Fig. 10, and in Fig. S6 in the supplemental material. Over Eastern China and other parts of the Asian continent, dry and wet deposition contributed almost equally to the total deposition of N, whereas dry deposition was dominant over Western China to the northwestern boundary of China. Over East Asia, from Korea to Japan, the simulated proportion of

wet deposition was higher, above 50%, using models M1, M2, M4, M5, M6, and M13, whereas models M11 and M12 showed predominantly wet deposition, above 70% and model M14 showed predominantly dry deposition over coastal areas. We note here that the uncertainty in anthropogenic NOx emissions over Asia has been reported to be about ±40% (Kurokawa et al., 2013). From the perspective of the balance between the deposition and emissions of N, it was illustrated that an excess of deposition was commonly shown over North Asia (Mongolia and Russia) and continental Southeast Asia (Myanmar, Lao

PDR, and Cambodia). Over these five countries, the amounts of deposition were greater than anthropogenic NOx emissions by at least 1.5 times. The importance of long-range transport from other countries in Asia, lateral boundaries, especially for North Asia, and biomass burning, especially for Southeast Asia, have been suggested. In other countries in Southeast Asia, namely Thailand, the Philippines, Malaysia, and Indonesia, a few models estimated a slight excess of N deposition compared with emissions, but most concluded that local sources were responsible for the total deposition of N. In Japan, model

discrepancies were large. Some previous studies have noted the influence of transboundary transport (Morino et al., 2011; Kajino et al., 2013; Itahashi et al., 2016), whereas other studies have estimated smaller impacts (Lin et al., 2008; Ge et al., 2014). Within the framework of MICS-Asia, further study would be needed to investigate the source–receptor relationships.

The analysis of the total deposition of A (Fig. 11) posed the combined results of the total depositions of S (Fig. 9) and N (Fig. 10) due to the ion counterpart. The simulated total deposition of A was highest over China, at 4000–8000 Gg N year$^{-1}$, followed

by Indonesia, at 900–1600 Gg N year$^{-1}$; for other countries, the deposition was below 1000 Gg N year$^{-1}$. Again, the deposition surpassed emissions in North Asia, and long-range transport within and outside of Asia was found to be important. Over Japan, model M12 showed a comparable proportion of deposition and emissions, while other models showed an excess of deposition over emissions. Over Southeast Asia excluding the Philippines, the possible importance of long-range transport for the total deposition of A is suggested by other results. The contribution of wet deposition to the total deposition of A is shown in Fig.

S9 in the supplemental material. The contrast between land and ocean was clearer compared with the proportions of the wet and dry depositions of S and N (Figs. S3 and S6). Over land, the proportion of wet deposition to the total deposition of A was either comparable to or more skewed toward dry deposition than what was found over India and some parts of China and over Southeast Asia. In contrast, over ocean, the proportion was above 80%, demonstrating the importance of the wet deposition of A. Models M11 and M13 showed a lower importance of wet deposition over ocean, with a proportion of 70% to total

deposition. Here, we remind the reader of the performance in Southeast Asia (Fig. 7). The analyzed models generally overestimated the wet deposition of A over Southeast Asia, and all models significantly overestimated it over the Philippines and Malaysia. By taking into account these model performances, the interpretation of the long-range transport effect will be changed. Compared with SO$_2$ and NOx emissions, which originate mainly from combustion processes, the uncertainty in NH$_3$



emissions is larger, being greater than ±100% (Kurokawa et al., 2013). Future studies should attempt to refine the emission inventory and understand the effect of emission uncertainties. The modeled total deposition of A was higher in India than in China. However, we cannot evaluate the model performance over India due to a lack of observation.

## 4.2 Ensemble approach

An ensemble approach was used with the aim of improving model performance. In MICS-Asia Phase II, it was found that the model ensemble means better agreed with measurements of sulfate and total ammonium than the individual results from each model (Hayami et al., 2008). Other model comparison studies, such as the Air Quality Model Evaluation International Initiative (AQMEII) over North America and Europe, also noted better model performance through the ensemble mean (Solazzo et al., 2012). In this MICS-Asia Phase III, other companion papers also tried to use the ensemble approach for the gas species $NO_2$,

$NH_3$, and CO (Kong et al., 2019), aerosols (Chen et al., 2019), and $O_3$ (Li et al., 2019). Here, we first used the following simple ensemble approach with all models:

$$\text{ENS} = \frac{1}{N} \sum D \qquad (4)$$

where D is the deposition (i.e., dry, wet, and total deposition), ENS is the ensemble mean, and N is the number of models; usually, N is 9, but can be 7 or 8 in models M13 and M14 due to the different model domain used. ENS and its coefficient of

variation (CV) were calculated, where CV is defined as the standard deviation divided by the mean; therefore, a large value of CV indicates inconsistency among models, while a small value indicates consistency among models. The ENS and CV of the wet depositions of S, N, and A are shown in Fig. 12, and those of the dry and total depositions of S, N, and A are respectively shown in Figs. S7 and S8. As shown in Fig. 12, the ENS for the wet deposition of S was high over China and Indonesia, that for the wet deposition of N was high over China, and that for the wet deposition of A was high over China, Indonesia, and

India, as was also simulated by each model (Figs. 4, 6, and 8). It was clarified that the CV values corresponding to these areas of high wet deposition were relatively small, with a similar result obtained in all models. These results were introduced in the performance for aerosol in MICS-Asia Phase III (Chen et al., 2019) and Phase II (Carmichael et al., 2008b). For example, over Eastern China, where a remarkably large amount of wet deposition of S and A was simulated, CV values varied from approximately 0.1 to 0.3. Due to differences in model performance, a slightly higher value of CV, around 0.4, was reported

for the wet deposition of N (see Section 3.2.2). Generally, CV values higher than 0.5 corresponded well to the area of small deposition amount, and CV values greater than 1.0 were found over Tibet, around Japan, and from Eastern Vietnam to Taiwan for the wet deposition of S; over the southwestern boundary of modeling domain for the wet deposition of N; and over the northwestern and southeastern boundaries of modeling domain for the wet deposition of A.



Another ensemble mean approach to emphasize model performance is the weighted ensemble mean (WENS). In MICS-Asia Phase II, the ratio of each R value to the sum was defined as the weighting factor (W) of the corresponding model (Wang et al., 2008). Here, we also applied R to derive the WENS as follows:

$$\text{WENS} = \frac{\sum(W \times D)}{\sum W} \qquad (5)$$

The results of the WENS are shown in Fig. 13. Overall, this approach gave results similar to those of ENS (shown in Fig. 12). The differences between ENS and WENS (ENS − WENS) were also calculated, and are shown in Fig. 13. For the wet deposition of S, positive differences (i.e., higher values estimated by ENS than WENS) were found around some parts of Eastern China, the Korean Peninsula, Japan and surrounding oceans, and Eastern Vietnam to Taiwan. For these areas, models simulated high amounts of wet deposition, and there were large differences among models. Over these areas, models M11 and
M12 simulated higher wet depositions than other models (see Fig. 4); however, values of R (i.e., weighting factors) were lower compared with other models (see Table 3). As a result, ENS led to higher wet deposition than WENS. Compared with the differences in the wet deposition of S between ENS and WENS, those of N and A were small; the differences were almost within ±1.0 kg N ha$^{-1}$ year$^{-1}$. This is because the values of R were similar among the models (see Tables 5 and 6), and almost the same estimation was obtained with ENS and WENS, even though there were large differences in the amount of wet
deposition.

The statistical performances obtained by ENS and WENS are listed in Tables 6 to 8. For the wet deposition of S, since only model M11 produced overestimation, ENS showed a negative NMB, and the R values of model M11 were lower compared with those of other models. WENS showed a larger NMB. In terms of NME, FAC2, and FAC3, ENS and WENS produced comparable results to those of model M13, which performed the best regarding NME, FAC2, and FAC3. For the wet deposition
of N, each model performed differently, and both ENS and WENS performed well in canceling the large outlier of each model performance. For the wet deposition of A, ENS and WENS also performed well in canceling the large outlier of each model performance. In terms of NME, model M14 performed best, and ENS, and especially WENS, showed comparable results. In summary, the ensemble and weighted ensemble approaches were confirmed to be better ways to improve the model performance, allowing the elimination of extreme performance.

**4.3 Precipitation-adjusted approach**

In Section 4.2, it was confirmed that the ensemble approach and the weighted ensemble approach, which considered R as a weighting factor, improved model performances for wet depositions and were effective at modulating the differences between models. Here, we sought a way to improve model performance and focused on the reproducibility of precipitation. The model performance for precipitation is clearly an important factor for wet deposition. In this MICS-Asia Phase III, the meteorological
field was basically coordinated in WRF, and three types of meteorological models were used (WRF, GEOS5, and RAMS).



Although the precipitation performances of all of the models generally captured the observed precipitation, their behaviors were different—for example, WRF and GEOS5 slightly overestimated and RAMS underestimated the precipitation, as discussed in Section 3.1. It is interesting that, compared with EANET observations, the model performances for the wet depositions of S, N, and A were found to have remarkably higher R values for models M13 and M14 driven by the different models of GEOS5 and RAMS. Such model performances for wet deposition might be partly due to the differences in precipitation. The precipitation-adjusted approach linearly scaled the precipitation amount to obtain the precipitation-adjusted wet deposition by the following equation:

$$\text{Adjusted } WD = \sum_{monthly} \text{Original } WD_{model} \times \frac{\sum_{monthly} P_{observation}}{\sum_{monthly} P_{model}} \qquad (6)$$

where $WD_{model}$ is the original modeled wet deposition and $P_{model}$ and $P_{observation}$ are the modeled and observed precipitation amounts, respectively. Note that this precipitation-adjusted approach assumes that errors associated with the modeled precipitation are linearly related to errors in wet deposition amounts. This approach has been used in previous studies over the USA (Appel et al., 2011; Zhang et al., 2018) and Asia (Itahashi, 2018). In this study, wet depositions were adjusted for precipitation on a monthly time scale, and then the annual precipitation-adjusted wet deposition was calculated. To verify this approach, soccer goal plots in terms of NMB and NME were created for the wet depositions of S, N, and A, as shown in Fig. 14. In these plots, R is indicated by the size of the circle.

For the wet deposition of S (Fig. 14 (a)), the improvement of model performance was clear; all of the model results were close to the soccer goal, and the size of the circle was larger. The model performance for the wet deposition of S improved values of R to above 0.7 for models M1, M2, M4, M5, and M6, above 0.5 for model M11, and above 0.6 for model M12; all of these models were driven by the WRF meteorological model. For model M13, which was driven by the GEOS5 model, an R value of 0.74 was obtained, and an R value of 0.64 was obtained for model M14, which was driven by the RAMS model. The underestimation, as shown by negative NMB, was improved by 10–20%, and NME was also improved by 10–20% for models M1, M2, M4, M5, M6, M13, and M14. The overestimation for model M11, as shown by positive NMB, was improved by 5%, and NME was improved by more than 20%. For model M12, NMB changed sign from negative bias to positive bias and NME was almost unchanged. We further conducted the ensemble approach for this precipitation-adjusted wet deposition. The statistical analysis is listed in Table 6. The ENS based on the precipitation-adjusted wet deposition showed an R value of 0.76, and both NMB and NME were improved compared with ENS based on the original wet deposition, also showing a better correspondence with observation; over 60% were within FAC2 and over 80% were within FAC3. The WENS based on the precipitation-adjusted approach also performed reasonably well and achieved a slightly worse NMB score, but other scores were almost the same for the ENS based on the precipitation-adjusted wet deposition.



For the wet deposition of N (Fig. 14 (b)), the results from the precipitation-adjusted approach were complicated. The CMAQ models M1, M2, M4, M5, and M6, which were driven by WRF, showed an improvement of R, however the values were different, ranging from 0.58 for M5 to 0.74 for M4. Since the wet deposition of N was differently calculated, even in CMAQ models, this precipitation-adjusted approach led to a better NMB for models M2, M4, and M6, whereas it led to a worse NMB for models M1 and M5; however, the latter two models showed almost no change in NME. Models M11 and M12, which were driven by WRF, did not show improvement in R through the precipitation-adjusted approach; the values of R were slightly reduced, and those of NME were reduced by more than 20%. The difference in performance between the original and precipitation-adjusted wet deposition was not dramatic for model M13, which was driven by GEOS5, and it revealed improvement for model M14, which was driven by RAMS, in terms of R, NMB, and NME. Thus, the use of ENS and WENS for the precipitation-adjusted approach listed in Table 7 achieves better performance for R and NMB, but no improvement in NME. Additionally, the corresponding percentages were improved.

For the wet deposition of A (Fig. 14 (c)), the precipitation-adjusted approach clearly improved the model performance. All model soccer goal plots were close to the center of the goal, and the sizes of the plots were enlarged; all models obtained R values of around 0.6. Generally, both NMB and NME were improved by about 10%, but model M12 changed to a slight overestimation. For model M13, which was driven by GEOS5, a reduction of overestimation was found. Therefore, the statistical performances of ENS and WENS, as listed in Table 8, suggest improvements of the original wet deposition simulations.

With the use of the precipitation-adjusted approach, overall improvements in model performances were shown, regardless of the original meteorological field. This result suggests the importance of the accuracy of modeled precipitation for the modeling of wet deposition. However, the precipitation-adjusted approach was not effective in terms of NME for the wet deposition of N. The mechanism of the wet deposition process should be further investigated in future research.

## 5. Concluding remarks and future perspectives for Phase IV

MICS-Asia Phase III was conducted in order to understand the current modeling capabilities in Asia. In this overview of deposition, simulations of deposition by nine models were analyzed. The modeled wet depositions of S, N, and A were evaluated by comparison with the wet deposition observed by EANET. Generally, the models can capture the observed wet deposition, albeit with underestimation for S and A and large variability among models for N. Then, we moved to a discussion of the importance of dry and wet depositions to total deposition, and presented maps of the total deposition over Asia. The balance between depositions and emissions was analyzed and revealed the possibility of a contribution from long-range transport. We also discussed ways to improve modeling results by taking an ensemble approach and a weighted ensemble approach using R as a weighting factor, and by using a precipitation-adjusted approach. Both approaches can successfully be applied to improve model performance.



In this overview paper, a model evaluation was conducted by comparison with EANET observations. Over China, which showed the highest amount of deposition over Asia, EANET data was available at eight sites in four regions. The available observations in China have been limited over the past decades; however, there are extensive observations to capture them. A detailed model inter-comparison over China based on the available observations will be reported in work following our

companion paper. Additionally, we limited the evaluation to wet deposition, due to the difficulty of measuring dry deposition, and relied on the model performance to determine atmospheric concentrations. In the EANET framework, an inferential method was used that utilizes multiple observed atmospheric concentrations and estimated dry deposition velocities. A detailed discussion of modeled dry deposition comparing the inferential method will be presented in our forthcoming companion research.

To further understand S, investigating the behavior of $Na^+$ as a sea-salt (ss) tracer would be valuable to separately analyze ss and non-ss (nss) $SO_4^{2-}$ concentration and deposition. This is especially important in coastal areas. In this Phase III study, large discrepancies were found over Japan for the wet deposition of S. This point should be considered in Phase IV. Moreover, the balance among S, N, and A should be further studied. Along with the drastic changes of emissions in China (Li et al., 2017b), it has been demonstrated that the key species in terms of acid deposition over East Asia has changed from S to N (Itahashi et

al., 2014, 2015, 2018). In this Phase III study, we conducted a full-year model simulation for 2010, and were able to estimate the annual accumulated deposition over Asia from multi-model inter-comparison for the first time. On the one hand, it is further necessary to conduct longer-term inter-comparison, since the meteorology (i.e., precipitation) has year-to-year variation, and it is not known whether the multi-model performance can capture such variation. One example is that wet deposition over Korea was higher than that over Japan in 2010 but this tendency was reversed in other years (EANET, 2016a,b).

On the other hand, it is also necessary to focus on case studies, such as severe rainfall events. Although we can provide an overview of the modeled deposition for annual accumulation from Phase III, we did not conduct a detailed analysis of model performance. Tshe use of different temporal coverages would be an potentially useful approach in Phase IV. Moreover, an analysis of precipitation type (convective and non-convective) would be interesting, and the updating of emissions from 2010 will also be required to account for the recent drastic changes over China (e.g., Li et al., 2018) and Southeast Asian countries.

Finally, adjustment for precipitation in Phase III revealed a potential way to improve the simulation of wet deposition. The model performance for precipitation and related parameters (e.g., water vapor mixing ratio) should be refined in Phase IV as the key input data to CTMs. This approach could constitute one of the methodologies in the Measurement–Model Fusion for Global Total Atmospheric Deposition (MMF-GTAD) project under the Global Atmosphere Watch (GAW) program of the World Meteorological Organization (WMO) (WMO GAW, 2017).



## Author contribution

SI led the deposition analysis group in MICS-Asia III, performed one of the model simulations, and prepared the manuscript with contributions from all co-authors. BG and KS are the members of the deposition analysis group in MICS-Asia III, and discussed the results with SI. JSF, XW, KY, TN, JL, BG, MK, HL, and MZ performed the model simulations and contributed to submit their simulated deposition results. ZW performed the meteorological model simulation and examined the model performance. ML and JK prepared the emission inventory data. MICS-Asia III was coordinated by GRC and ZW.

## Competing interests

The authors declare that they have no conflict of interest.

## Acknowledgement

The authors thank EANET for providing wet deposition measurement data over Asia. The authors are grateful to Dr. Kengo Sudo for providing us with the CHASER data as the lateral boundary condition. Syuichi Itahashi acknowledges the support of JSPS KAKENHI (Grant JP16K21690).

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



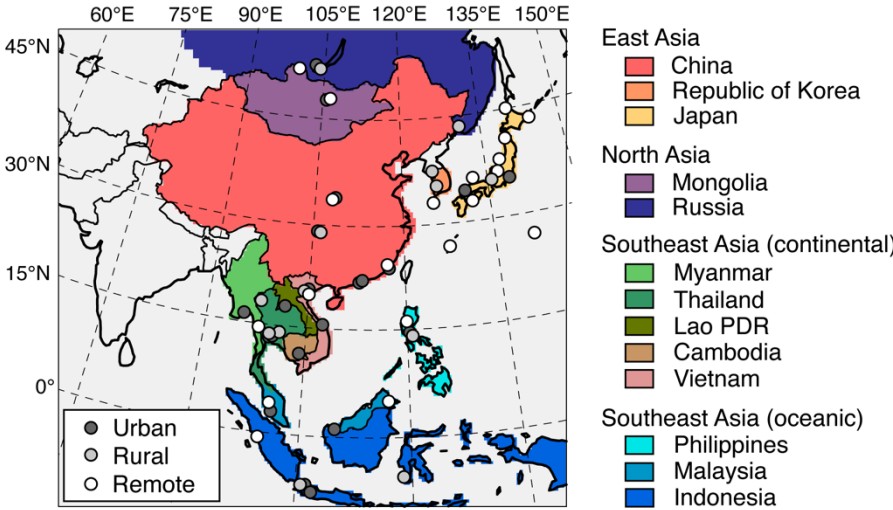

**Figure 1. Map of the unified domain of the Model Inter-Comparison Study for Asia (MICS-Asia) Phase III. Circles with different colors indicate observation sites in remote (white), rural (light gray), and urban (dark gray) areas as classified by the Acid Deposition Monitoring Network in East Asia (EANET) definitions. Map colors indicate the 13 countries participating in EANET in 2010 that were used for the analysis of deposition amount, which were classified into four regions in this study.**





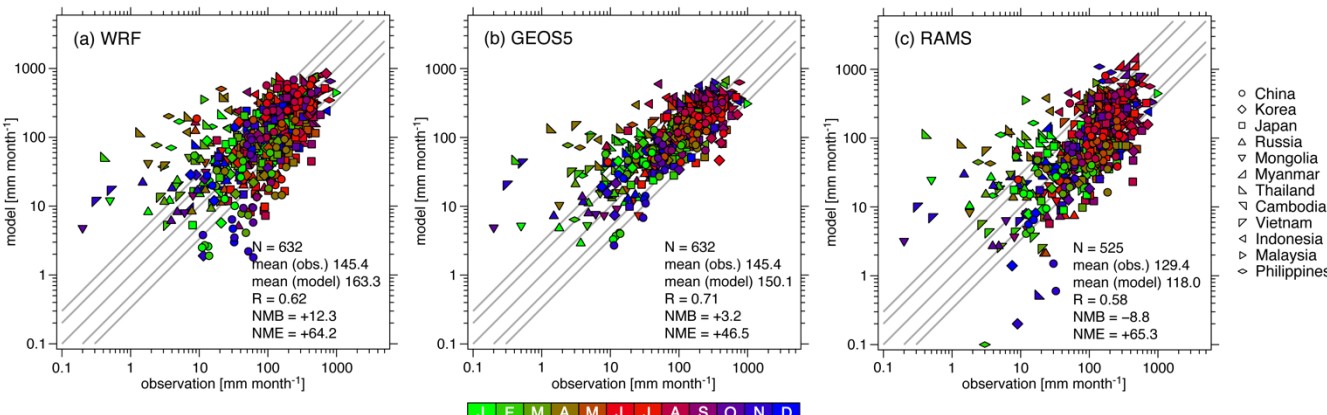

**Figure 2. Scatter plots of monthly precipitation amount over Asia simulated by (a) the Weather Research and Forecasting (WRF) model, (b) the Goddard Earth Observing System 5 (GEOS5) model, and (c) the Regional Atmospheric Modeling System (RAMS) model. Symbols indicate different countries and colors indicate different months. Statistical analysis of the mean, correlation coefficient (R), normalized mean bias (NMB), and normalized mean error (NME) is shown in the inset.**





**Figure 3. Model evaluation of the annual accumulated wet deposition of sulfate aerosol, sulfur dioxide (SO₂), and sulfuric acid (H₂SO₄) (S) (a) at each EANET observation site, (b) for countries, (c) over four defined regions, and (d) over different site classifications.**



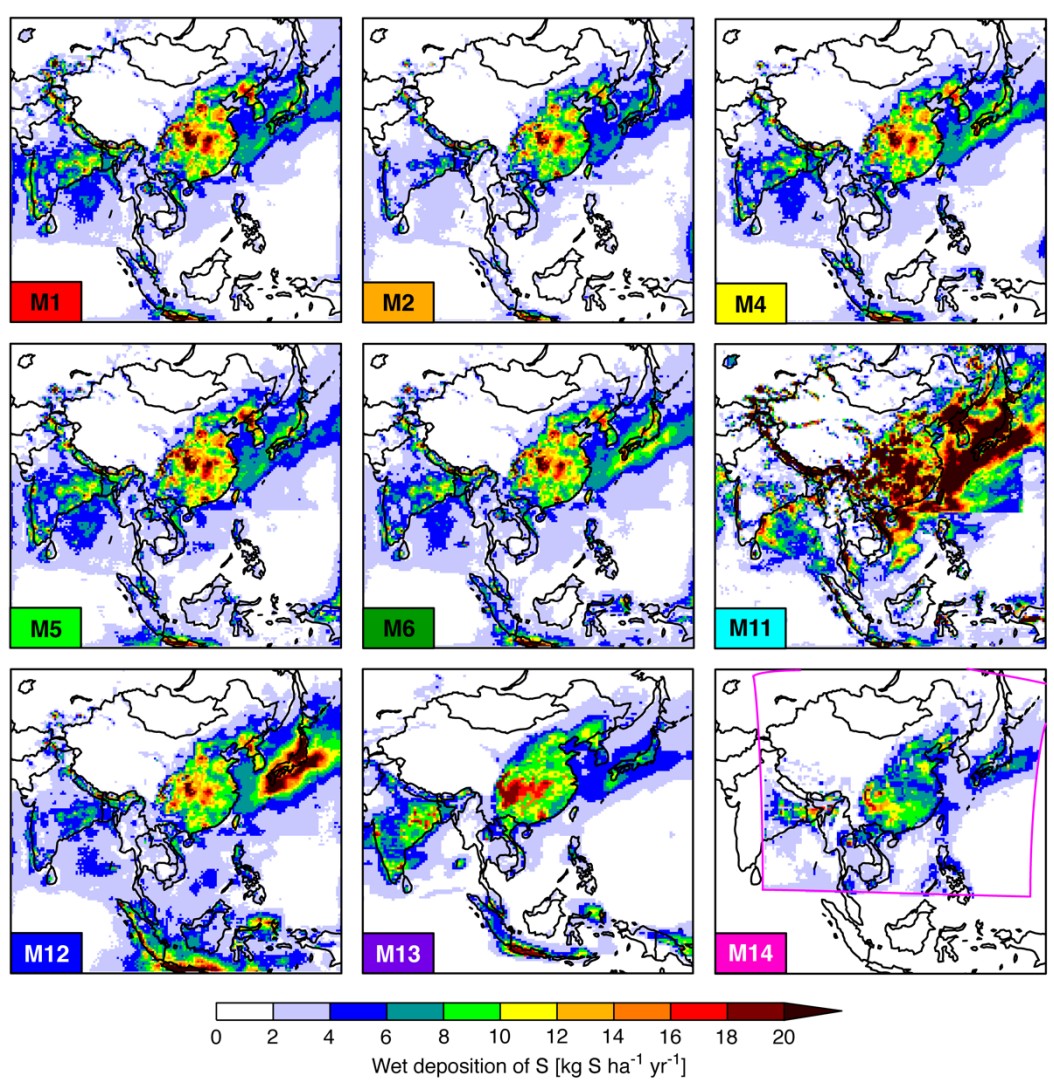

**Figure 4. Spatial distributions of the annual accumulated wet deposition of S.**







**Figure 5.** Model evaluation of the annual accumulated wet deposition of nitrate aerosol, nitrogen monoxide (NO), nitrogen dioxide (NO₂), and nitric acid (HNO₃) (N) (a) at each EANET observation site, (b) for countries, (c) over four defined regions, and (d) over different site classifications.





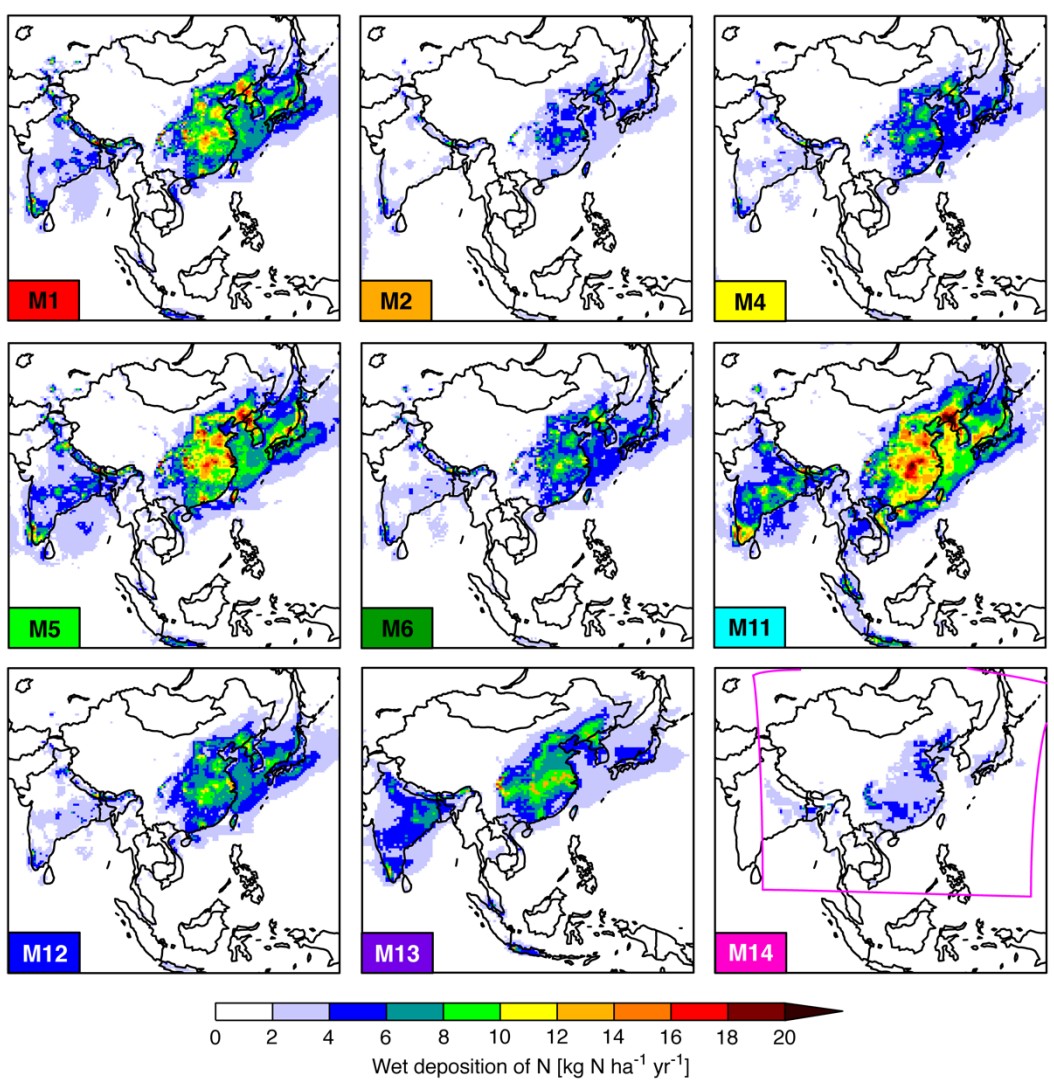

**Figure 6. Spatial distributions of the annual accumulated wet deposition of N.**





**Figure 7. Model evaluation of the annual accumulated wet deposition of ammonium aerosol and ammonia (NH₃) (A) (a) at each EANET observation site, (b) for countries, (c) over four defined regions, and (d) over different site classifications.**





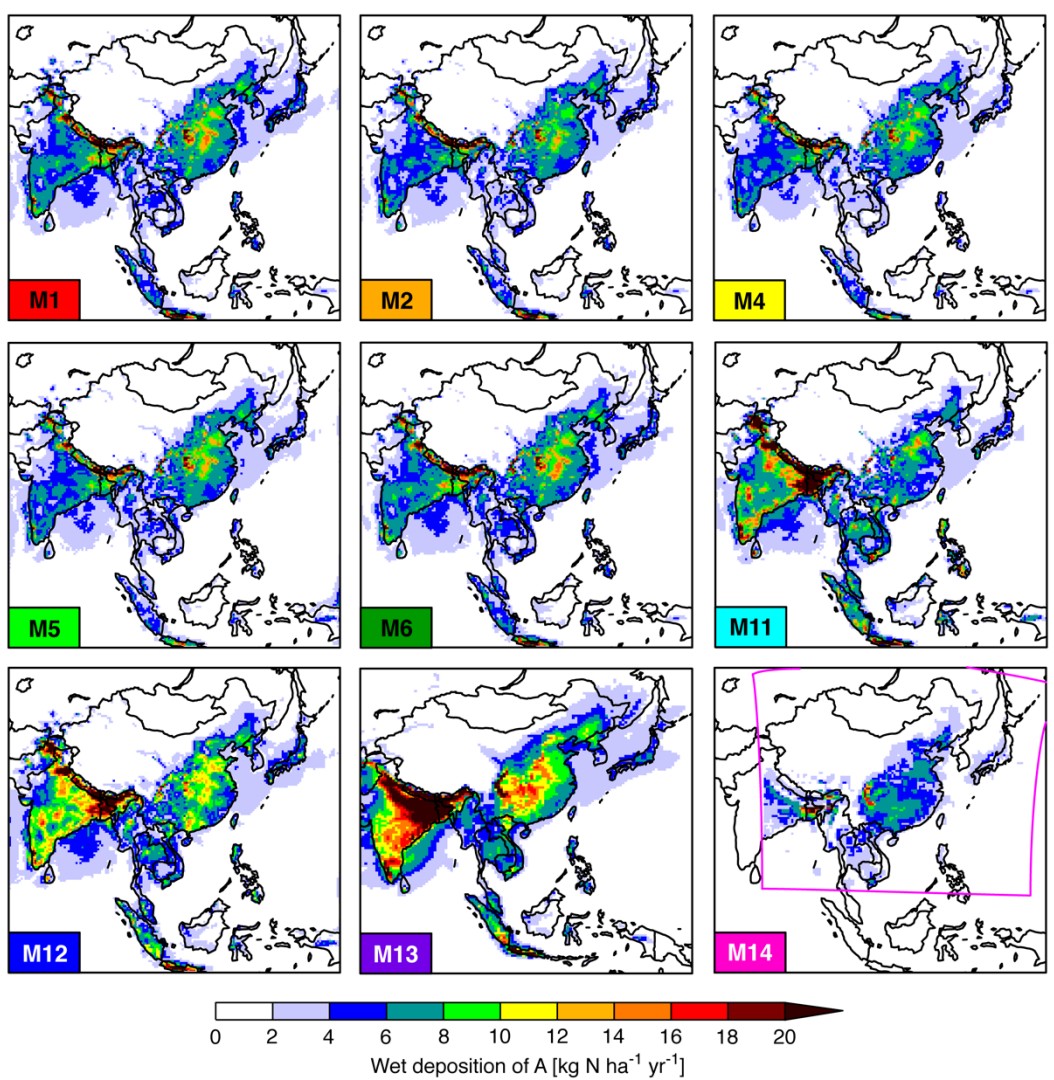

**Figure 8. Spatial distributions of the annual accumulated wet deposition of A.**





**Figure 9. Map of the annual accumulated total S deposition over Asia. The cumulative bar graphs show dry deposition (light colors) and wet deposition (dark colors), with different colors representing different models and black representing the amount of anthropogenic SO₂ emissions. The background red areas of the graphs indicate an excess of total depositions compared to emissions, and their transparency is based on the number of models making this prediction (i.e., bolder colors indicate that more of the nine models simulated the excess).**





**Figure 10. Map of the annual accumulated total N deposition over Asia. The cumulative bar graphs show dry deposition (light colors) and wet deposition (dark colors), with different colors representing different models and black representing the amount of anthropogenic NOx emissions. The background blue areas of the graphs indicate an excess of total depositions compared to emissions, and their transparency is based on the number of models making this prediction (i.e., bolder colors indicate that more of the nine models simulated the excess).**




**Figure 11. Map of the annual accumulated total A deposition over Asia. The cumulative bar graphs show dry deposition (light colors) and wet deposition (dark colors), with different colors representing different models and black representing the amount of anthropogenic NH₃ emissions. The background green areas of the graphs indicate the excess of total depositions compared to emissions, and their transparency is based on the number of models making this prediction (i.e., bolder colors indicate that more of the nine models simulated the excess).**





**Figure 12. Spatial distributions of the ensemble mean (left) and the coefficient of variation (right) for the wet deposition of (a) S, (b) N, and (c) A.**



**Figure 13. (Left) Spatial distributions of the weighted ensemble mean for the wet deposition of (a) S, (b) N, and (c) A. (Right) Differences between the ensemble mean and the weighted ensemble mean (calculated as ensemble mean – weighted ensemble mean).**





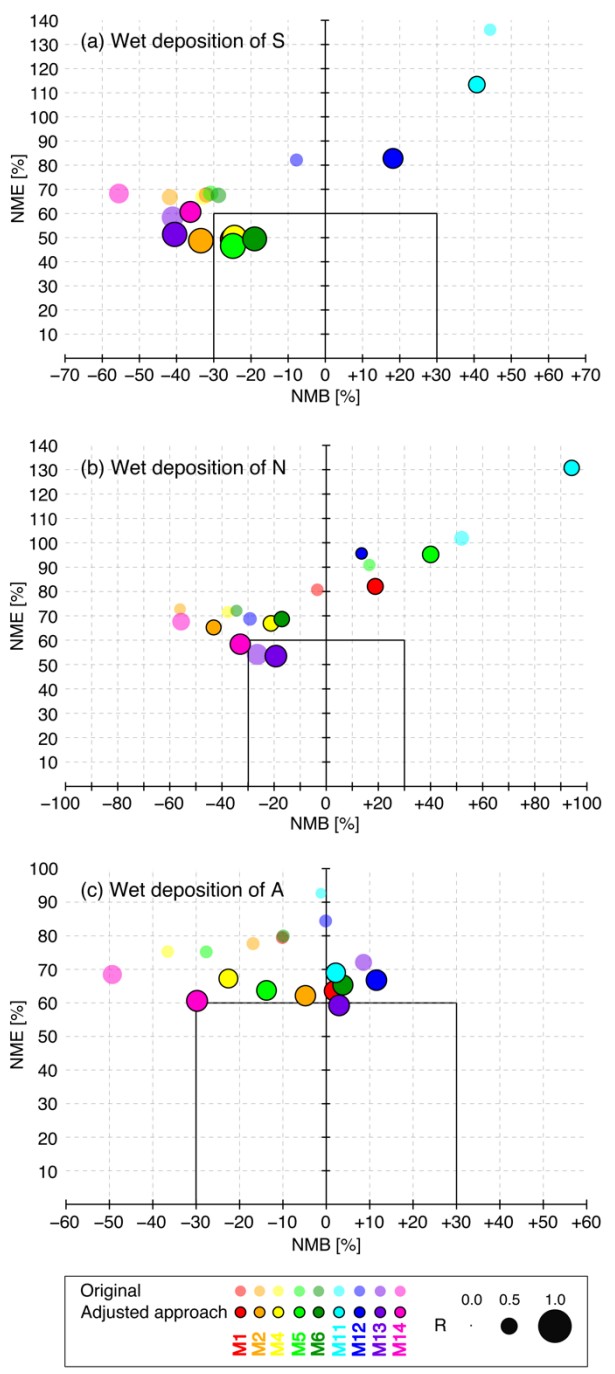

**Figure 14. Soccer-goal plot of NMB (x-axis) and NME (y-axis) for the original wet deposition (transparent circles) and precipitation-adjusted wet deposition (solid circles) of (a) S, (b) N, and (c) A. The size of each circle indicates R. Note that the ranges of NMB and NME are different among the three panels.**





**Table 1. Descriptions of the models used in this acid deposition study.**

| No. | M1 | M2 | M4 | M5 | M6 | M11 | M12 | M13 | M14 |
|---|---|---|---|---|---|---|---|---|---|
| Model (Version) | CMAQ (5.0.2) | CMAQ (5.0.2) | CMAQ (4.7.1) | CMAQ (4.7.1) | CMAQ (4.7.1) | NAQPMS | NHM Chem | Geos-Chem (9.1.3) | CMAQ (4.6) |
| Domain[a] | Standard | Standard | Standard | Standard | Standard | Standard | Standard | Nested Asia | Smaller area in Asia |
| Meteorological field | WRF | WRF | WRF | WRF | WRF | WRF | WRF | GEOS-5 | RAMS |
| Horizontal advection[b] | Yamo | Yamo | PPM | PPM | Yamo | WA | WA | TPCORE | PPM |
| Vertical advection[b] | PPM | PPM | PPM | PPM | Yamo | WA | WA | TPCORE | PPM |
| Horizontal diffusion[c] | multiscale | multiscale | multiscale | multiscale | multiscale | BD | multiscale | HB | multiscale |
| Vertical diffusion[c] | ACM2 | ACM2 | ACM2 (inline) | ACM2 | ACM2 (inline) | BD | MYJ | HB | ACM2 |
| Gas-phase chemistry[d] | SAPRC-99 | SAPRC-99 | SAPRC-99 | SAPRC-99 | SAPRC-99 | CBMZ | SAPRC-99 | Bey | SAPRC-99 |
| Aerosol chemistry[e] | AERO6 | AERO6 | AERO5 | AERO5 | AERO5 | Li | Kajino | Park, Pye | AERO5 |
| Thermodynamics[f] | version 2.1 | version 2.1 | version 1.7 | version 1.7 | version 1.7 | version 1.7 | version 2.1 | version 2.1 | version 1.7 |
| Dry deposition[g] | M3DRY | M3DRY | M3DRY | M3DRY | M3DRY | Wesely | Kajino | Wesely, Wang | M3DRY |
| Surface layer height | 58 m | 58 m | 58 m | 58 m | 58 m | 48 m | 27 m | 123 m | 100 m |
| Wet deposition[h] | Foley | Foley | Foley | Foley | Foley | Ge | Kajino | Liu | Foley |
| Boundary condition[i] | GEOS-Chem | Default | CHASER | CHASER | CHASER | CHASER | CHASER | GEOS-Chem | GEOS-Chem |

a: "Standard" indicates the unified domain in the Model Inter-Comparison Study for Asia (MICS-Asia) Phase III. See text for details.

b: References for the advection scheme are as follows: Yamo: Yamartino, 1993; PPM: Piecewise Parabolic Method (Colella and Woodward, 1984); WA: Walcek and Aleksic, 1998; TPCORE: Wang et al., 2004.

c: References for the diffusion scheme are as follows: ACM2: Asymmetric Convective Model version 2 (Pleim, 2007a,b); BD: Byun and Dennis, 1995; HB: Holtslag and Boville, 1993; multiscale: Byun and Schere, 2006; MYJ: Janjic, 1994.

d: References for the gas-phase chemistry are as follows: Bey: Bey et al., 2001; CBMZ: Zaveri and Peters, 1999; SAPRC-99: Carter, 2000.

e: References for the aerosol chemistry are as follows: AERO5, AERO6, Kajino: Kajino et al., 2018; Li: Li et al., 2011; Park: Park, 2004; Pye: Pye et al., 2009.



f: On thermodynamics. All models use ISORROPIA but different versions, namely version 1.7 (Nenes et al., 1998) or version 2.1 (Fountoukis and Nenes, 2008).

g: References for the dry deposition scheme are as follows: M3DRY: Pleim et al., 2001; Kajino: Kajino et al., 2018; Wang: Wang et al., 2004; Wesely: Wesely, 1989.

5  h: References for the wet deposition scheme are as follows: Foley: Foley et al., 2010; Ge: Ge et al., 2014; Kajino: Kajino et al., 2018; Liu: Liu et al., 2001.

i: References for the boundary condition are as follows: CHASER: Sudo et al., 2002a,b; GEOS-Chem: Bey et al., 2001. Note that model M2 adopted the default boundary condition in the Community Multiscale Air Quality (CMAQ) modeling system.

**Table 2**. Locations and methodology for 54 Acid Deposition Monitoring Network in East Asia (EANET) observation sites.





| No. | Country | Name | Latitude (°) | Longitude (°E) | Altitude (m a.s.l.) | Classification | Sampling interval | Anion analysis | Cation analysis |
|---|---|---|---|---|---|---|---|---|---|
| 1 | China | Zhuxiandong | 22.20 | 113.52 | 45 | Urban | Daily | IC | IC |
| 2 | | Xiangzhou | 22.27 | 113.57 | 40 | Urban | Daily | IC | IC |
| 3 | | Hongwen | 24.47 | 118.13 | 50 | Urban | Daily | IC | IC |
| 4 | | Xiaoping | 24.85 | 118.03 | 686 | Remote | Daily | IC | IC |
| 5 | | Haifu | 29.62 | 106.50 | 317 | Urban | Daily | IC | IC |
| 6 | | Jinyunshan | 29.82 | 106.37 | 262 | Rural | Daily | IC | IC |
| 7 | | Shizhan | 34.23 | 108.95 | 400 | Urban | Daily | IC | IC |
| 8 | | Jiwozi | 33.83 | 108.80 | 1800 | Remote | Daily | IC | IC |
| 9 | Republic of Korea | Cheju | 33.30 | 126.16 | 72 | Remote | Daily | IC | IC |
| 10 | | Imsil | 35.60 | 127.18 | | Rural | Daily | IC | IC |
| 11 | | Kanghwha | 37.70 | 126.28 | 150 | Rural | Daily | IC | IC |
| 12 | Japan | Hedo | 26.87 | 128.25 | 60 | Remote | Daily | IC | IC |
| 13 | | Ogasawara | 27.09 | 142.22 | 230 | Remote | Daily | IC | IC |
| 14 | | Yusuhara | 33.38 | 132.93 | 790 | Remote | Daily | IC | IC |
| 15 | | Banryu | 34.68 | 131.80 | 53 | Urban | Weekly | IC | IC |
| 16 | | Oki | 36.29 | 133.19 | 90 | Remote | Daily | IC | IC |
| 17 | | Ijira | 35.57 | 136.69 | 140 | Rural | Weekly | IC | IC |
| 18 | | Tokyo | 35.69 | 139.76 | 26 | Urban | Daily | IC | IC |
| 19 | | Happo | 36.70 | 137.80 | 1850 | Remote | Daily | IC | IC |
| 20 | | Sadoseki | 38.25 | 138.40 | 136 | Remote | Daily | IC | IC |
| 21 | | Tappi | 41.25 | 140.35 | 106 | Remote | Daily | IC | IC |
| 22 | | Rishiri | 45.12 | 141.21 | 40 | Remote | Daily | IC | IC |
| 23 | | Ochiishi | 43.16 | 145.50 | 49 | Remote | Daily | IC | IC |
| 24 | Mongolia | Ulaanbaatar | 47.90 | 106.82 | 1282 | Urban | Daily | IC | IC |
| 25 | | Terelj | 47.98 | 107.48 | 1540 | Remote | Daily | IC | IC |
| 26 | Russia | Mondy | 51.67 | 101.00 | 2000 | Remote | Daily | IC | SP |
| 27 | | Listvyanka | 51.85 | 104.90 | 700 | Rural | Daily | IC | SP |
| 28 | | Irkutsk | 52.23 | 104.25 | 400 | Urban | Daily | IC | SP |
| 29 | | Primorskaya | 43.70 | 132.12 | 84 | Rural | Daily | SP, TI, NP | SP |





| 30 | Myanmar | Yangon | 16.50 | 96.12 | 22 | Urban | Daily | IC | IC |
|---|---|---|---|---|---|---|---|---|---|
| 31 | Thailand | Bangkok | 13.77 | 100.53 | 2 | Urban | Daily | IC | IC |
| 32 | | Samutprakarn | 13.73 | 100.57 | 2 | Urban | Daily | IC | IC |
| 33 | | Pathumthani | 14.03 | 100.77 | 2 | Rural | Daily | IC | IC |
| 34 | | Khanchanaburi | 14.77 | 98.58 | 170 | Remote | Daily | IC | IC |
| 35 | | Nakhon Ratchasima | 14.45 | 101.88 | 418 | Rural | Daily | IC | IC |
| 36 | | Chiang Mai | 18.77 | 98.93 | 350 | Rural | Daily | IC | IC |
| 37 | Lao PDR | Vientiane | 17.00 | 102.00 | | Urban | Daily | IC | IC |
| 38 | Cambodia | Phnom Penh | 11.55 | 104.83 | 10 | Urban | Weekly | IC | IC |
| 39 | Vietnam | Da Nang | 16.04 | 108.21 | 60 | Urban | 10 days | IC | IC |
| 40 | | Hanoi | 21.02 | 105.85 | 5 | Urban | Weekly | IC | IC |
| 41 | | Hoa Binh | 20.82 | 105.33 | 23 | Rural | Weekly | IC | IC |
| 42 | | Cuc Phuong | 20.25 | 105.72 | 155 | Remote | 10 days | IC | IC |
| 43 | Philippines | Metro Manila | 14,63 | 121.07 | 54 | Urban | Weekly | IC | IC |
| 44 | | Los Banos | 14.18 | 121.25 | 35 | Rural | Weekly | IC | IC |
| 45 | | Mt. Sto. Tomas | 16.42 | 120.60 | 1500 | Rural | Weekly | IC | IC |
| 46 | Malaysia | Petaling Jaya | 3.10 | 101.65 | 87 | Urban | Weekly | IC | IC |
| 47 | | Tanah Rata | 4.47 | 101.38 | 1470 | Remote | Weekly | IC | IC |
| 48 | | Kuching | 1.48 | 110.47 | 22 | Urban | Weekly | IC | IC |
| 49 | | Danum Valley | 4.98 | 117.85 | 427 | Remote | Weekly | IC | IC |
| 50 | Indonesia | Kototabang | −0.20 | 100.32 | 864 | Remote | Weekly | IC | IC |
| 51 | | Jakarta | −6.18 | 106.83 | 7 | Urban | Weekly | IC | IC |
| 52 | | Bandung | −6.90 | 107.58 | 743 | Urban | Daily | IC | IC |
| 53 | | Serpong | −6.25 | 106.57 | 46 | Rural | Daily | IC | IC |
| 54 | | Maros | −4.92 | 119.57 | 11 | Rural | Weekly | IC | IC |

IC: ion chromatography; SP: spectrophotometry; TI: titration; NP: nephelometry; PDR: People's Democratic Republic.



**Table 3. Summary of statistical analysis of model performance for the wet deposition of S.**

| Model | M1 | M2 | M4 | M5 | M6 | M11 | M12 | M13 | M14 |
|---|---|---|---|---|---|---|---|---|---|
| N | | | | 588 | | | | | 492 |
| mean (observation) [g S ha$^{-1}$ month$^{-1}$] | | | | 876.5 | | | | | 868.0 |
| mean (model) [g S ha$^{-1}$ month$^{-1}$] | 596.6 | 510.1 | 583.0 | 606.4 | 624.5 | 1264.9 | 808.4 | 516.6 | 386.1 |
| R | 0.43 | 0.46 | 0.44 | 0.43 | 0.44 | 0.34 | 0.35 | 0.63 | 0.56 |
| NMB [%] | −31.9 | −41.8 | −33.5 | −30.8 | −28.8 | +44.3 | −7.8 | −41.1 | −55.5 |
| NME [%] | +67.7 | +66.7 | +66.9 | +68.4 | +67.5 | +136.1 | +82.1 | +58.4 | +68.3 |
| FAC2 [%] | 40.0 | 39.3 | 41.0 | 40.1 | 42.3 | 29.8 | 39.6 | 44.9 | 38.6 |
| FAC3 [%] | 60.5 | 61.4 | 60.7 | 60.2 | 62.6 | 42.5 | 59.0 | 64.8 | 55.1 |

Note: Models M1, M2, M4, M5, M6, M11, and M12 were based on the unified meteorological field of WRF. Model M13 was based on the different meteorological model of GEOS5, and the covering domain was also different (see Fig. 4). Model M14 was based on the different meteorological model of RAMS, and the covering domain was also smaller (see Fig. 4).



**Table 4. Summary of statistical analysis of model performance for the wet deposition of N.**

| Model | M1 | M2 | M4 | M5 | M6 | M11 | M12 | M13 | M14 |
|---|---|---|---|---|---|---|---|---|---|
| N | | | | 575 | | | | | 482 |
| mean (observation) [g-N ha⁻¹ month⁻¹] | | | | 347.5 | | | | | 315.2 |
| mean (model) [g-N ha⁻¹ month⁻¹] | 337.5 | 153.4 | 217.4 | 407.2 | 229.1 | 530.8 | 247.1 | 256.8 | 140.6 |
| R | 0.35 | 0.32 | 0.34 | 0.34 | 0.33 | 0.42 | 0.38 | 0.62 | 0.51 |
| NMB [%] | −3.4 | −56.1 | −37.8 | +16.5 | −34.4 | +51.9 | −29.3 | −26.5 | −55.7 |
| NME [%] | +80.7 | +72.7 | +71.5 | +90.9 | +72.1 | +101.9 | +68.8 | +54.2 | +67.6 |
| FAC2 [%] | 42.8 | 31.1 | 39.1 | 41.4 | 40.9 | 44.2 | 43.8 | 54.3 | 34.9 |
| FAC3 [%] | 62.3 | 50.3 | 56.0 | 59.0 | 58.1 | 61.9 | 61.4 | 70.6 | 51.0 |

Note: Models M1, M2, M4, M5, M6, M11, and M12 were based on the unified meteorological field of WRF. Model M13 was based on the different meteorological model of GEOS5, and the covering domain was also different (see Fig. 6). Model M14 was based on the different meteorological model of RAMS, and the covering domain was also smaller (see Fig. 6).



**Table 5. Summary of statistical analysis of model performance for the wet deposition of A.**

| Model | M1 | M2 | M4 | M5 | M6 | M11 | M12 | M13 | M14 |
|---|---|---|---|---|---|---|---|---|---|
| N | | | | | 568 | | | | 474 |
| mean (observation) [g-N ha$^{-1}$ month$^{-1}$] | | | | | 440.7 | | | | 447.0 |
| mean (model) [g-N ha$^{-1}$ month$^{-1}$] | 394.7 | 365.0 | 278.5 | 317.7 | 395.5 | 433.7 | 438.7 | 476.9 | 225.6 |
| R | 0.35 | 0.36 | 0.34 | 0.35 | 0.35 | 0.28 | 0.35 | 0.48 | 0.54 |
| NMB [%] | −10.2 | −16.9 | −36.6 | −27.7 | −10.0 | −1.3 | −0.2 | +8.6 | −49.3 |
| NME [%] | +79.5 | +77.7 | +75.3 | +75.2 | +79.9 | +92.7 | +84.4 | +72.1 | +68.5 |
| FAC2 [%] | 39.6 | 37.7 | 35.4 | 38.4 | 38.2 | 31.9 | 35.9 | 50.9 | 35.0 |
| FAC3 [%] | 58.5 | 57.9 | 53.5 | 57.6 | 58.6 | 48.8 | 55.5 | 68.3 | 54.9 |

Note: Models M1, M2, M4, M5, M6, M11, and M12 were based on the unified meteorological field of WRF. Model M13 was based on the different meteorological model of GEOS5, and the covering domain was also different (see Fig. 8). Model M14 was based on the different meteorological model of RAMS, and the covering domain was also smaller (see Fig. 8).



**Table 6. Summary of statistical analysis of model performance by the ensemble and precipitation-adjusted approaches for the wet deposition of S.**

| Model | Ensemble mean | Weighted ensemble mean | Ensemble mean of precipitation-adjusted wet deposition | Weighted ensemble mean of precipitation-adjusted wet deposition |
|---|---|---|---|---|
| N | | | 588 | |
| mean (observation) [g S ha$^{-1}$ month$^{-1}$] | | | 876.5 | |
| mean (model) [g S ha$^{-1}$ month$^{-1}$] | 675.4 | 615.9 | 739.4 | 649.5 |
| R | 0.47 | 0.51 | 0.76 | 0.76 |
| NMB [%] | −22.9 | −29.7 | −15.6 | −19.9 |
| NME [%] | +66.5 | +62.8 | +48.4 | +47.1 |
| FAC2 [%] | 45.1 | 45.9 | 62.9 | 62.9 |
| FAC3 [%] | 63.1 | 63.8 | 81.6 | 81.8 |



**Table 7. A summary of the statistical analysis of model performance by ensemble and precipitation-adjusted approaches for the wet deposition of N.**

| Model | Ensemble mean | Weighted ensemble mean | Ensemble mean of precipitation-adjusted wet deposition | Weighted ensemble mean of precipitation-adjusted wet deposition |
|---|---|---|---|---|
| N | | | 575 | |
| mean (observation) [g N ha$^{-1}$ month$^{-1}$] | | | 347.5 | |
| mean (model) [g N ha$^{-1}$ month$^{-1}$] | 282.6 | 271.2 | 359.0 | 347.1 |
| R | 0.43 | 0.44 | 0.52 | 0.53 |
| NMB [%] | −19.1 | −22.4 | +3.2 | −0.3 |
| NME [%] | +67.2 | +65.0 | +68.3 | +66.4 |
| FAC2 [%] | 46.8 | 47.5 | 52.9 | 53.6 |
| FAC3 [%] | 66.3 | 68.0 | 74.8 | 73.9 |



**Table 8. A summary of the statistical analysis of model performance by ensemble and precipitation-adjusted approaches for the wet deposition of A.**

| Model | Ensemble mean | Weighted ensemble mean | Ensemble mean of precipitation-adjusted wet deposition | Weighted ensemble mean of precipitation-adjusted wet deposition |
|---|---|---|---|---|
| N | | | 568 | |
| mean (observation) [g N ha$^{-1}$ month$^{-1}$] | | | 440.7 | |
| mean (model) [g N ha$^{-1}$ month$^{-1}$] | 378.3 | 358.5 | 420.2 | 411.8 |
| R | 0.41 | 0.45 | 0.66 | 0.66 |
| NMB [%] | −13.6 | −18.4 | −4.6 | −6.6 |
| NME [%] | +74.6 | +70.8 | +58.0 | +57.6 |
| FAC2 [%] | 38.0 | 41.4 | 57.4 | 57.6 |
| FAC3 [%] | 59.5 | 61.4 | 76.4 | 76.9 |