# Peer review of "MICS-Asia III: Overview of model inter-comparison and evaluation of acid deposition over Asia"

_Atmospheric Chemistry and Physics, 2019_

## Referee Comment (RC1) · Anonymous Referee #1 · 1 Oct 2019

General Comments: Authors compare nine meteorology-chemical transport model systems to estimate deposition amount of Sulfur, Nitrogen, and Ammonia. In general, this manuscript is well organized and delivers informative results and an interesting air quality issue over the Northeast Asia. For the model performance evaluations, authors compare the model outputs to the EANET observations. It is reasonable. However, those nine models may already present appreciable differences in airborne concentrations (authors address the companion papers in the text, but short discussion would be helpful for other readers), or may estimate different wet deposition amounts even under the same atmospheric concentrations due to difference in the implemented dry deposition mechanism in the models. Therefore, it would be better to explain more direct relationship between concentrations and dry/wet depositions for model inter-comparisons.

[Figure]

Authors evaluate the precipitation simulations, but wet deposition in CMAQ (6 out of 9 models) not only depends on the precipitation but the types (convective or non-convective). Moreover, water mixing ratio also plays an important role in CMAQ depending on the meteorological conditions. Those analyses will be very helpful to the air quality research community.

Scientific Comments: Table 1: It would be better to include the WRF configurations. Physical options in WRF may affect the wet depositions of those target species.

Figure 2: Wet scavenging is affected by not only the precipitation amounts but also precipitation intensity (and the lasting time), and types (at least in CMAQ parameterization). For example, convective precipitation in MCIP outputs for CMAQ may increase the total rain amounts in summer but may have less influence on fine particle removal, compared to non-convective rain during spring and fall. In case of wet removals of gases species, total surface areas of rain droplets as function of droplet size and ambient bulk concentrations would be important. Size distributions and concentrations of ambient aerosol would be critical for wet scavenging.

Figure 3: Wet depositions in current three dimensional grid models (CMAQ and CAMx) deal with both in-cloud and below-cloud scavenging together. It would be okay to explain total wet depositions, but may mislead to the comparisons to the EANET measurements.

Figures 3~8: In terms of wet deposition comparisons, Figures and their explanations are well represented in the text. Considering that wet scavenging amounts are determined by airborne concentrations and removal mechanism, it is expected to relate modeled concentrations and wet deposition amounts, including the removal module used in the models.

---

## Referee Comment (RC2) · Anonymous Referee #2 · 1 Oct 2019

Review of MICS-Asia III: Overview of model inter-comparison and evaluation of acid deposition over Asia by Itahashi et al.

The paper deals with an intermodel comparison of wet deposition patterns for East Asia given present day conditions. The models used in this study are chemistry transport models and meteorological data is provided by external model fields (for most models form WRF). In general this study is worth publication, however the gain of new scientific knowledge from this effort is rather small. Consequently, would have proposed that this manuscript is much better suited for GMD instead of ACP as a typical evaluation paper. The paper is decently structured and despite some minor required language improvements (proof-reading) could be easily published. The model and observation data description sections are adequate; however if it might be possible to add the

observation data file in the supplement more model applicants would benefit from this study.

Despite the model domain is very similar (if not identical) in the simulations and the models use unified emissions, substantial differences in the deposition maps occur. Unfortunately, the authors do not show a comparison of the simulated concentrations or tropospheric vertical columns of the species the study focusses on. Therefore, it is difficult to judge, whether the differences occur from slightly different assumptions in the wet deposition schemes, the simulated precipitation amount or the simulated concentrations of the trace species and their precursors. As the applied chemistry and aerosol schemes differ to a certain degree, this could already be a major cause for the differences in the wet deposition patterns.

The evaluation of the precipitation is unfortunately only superficial. As only monthly mean precipitation is compared with the simulation results, the corresponding frequency distribution, i.e. the number, duration and intensity of the events cannot be determined. However, this is crucial for wet deposition, as a few short but intense rain events result in less deposition compared to longer precipitation events of average intensity. This will also substantially impact the precipitation adjustment (see comment below!).

Analysing the wet deposition of sulphur, M11 shows a substantially higher deposition pattern in China. What is the reason for this? This is a typical example for a model inter-comparison study where data is compared, but the causes for the differences are not analysed in detail. Has there been an issue with SO2 emissions or conversion from S(IV) to S(VI)? Is there a bias originating from seasalt sulphate? Is total sulphur completely overestimated in this model? Or is it completely depleted, as wet deposition is so efficient? These differences require much more analysis for a consistent inter-comparison study.

Especially, when creating ensembles including such outliers, the ensemble mean can

even be deteriorated compared to individual simulations. This does not appear to be the case in this study, as the M11 simulation compensates some of the low bias from the majority of the other simulations. A similar behaviour of overestimation is not as obvious for nitrate and ammonium, leading to the impression that this is not necessarily a consequence of the wet deposition scheme.

The weighted ensemble might be a better option to reduce the importance of outliers; however, it simply states that the models which show best agreement with the observations should be used for the ensemble mean. Consequently, it reduces ensemble spread and therefore does not cover the whole range of simulation results properly. Please state explicitly, what you hope to gain from the weighted ensemble mean.

Concerning the total deposition maps, the authors should clearly point out, that underestimated wet deposition can often be compensated by overestimated dry deposition and vice versa, as both processes depend on the atmospheric burden (or near surface concentrations).

I (personally) see the option of precipitation adjustment to improve the consistency of the simulation results with observations very critical. This adjustment does not include any kind of frequency distribution of precipitation events, the vertical extent of the precipitation (and hence the accessible fraction of the tracer vertical column for wet deposition). Also it does not include any kind of vertical redistribution by scavenging and subsequent evaporating precipitation and hence tracer release at lower altitude. Of course, I agree that with wrong precipitation amounts it will be impossible to fully match observations, but in my opinion not only the total amount of precipitation, but at least the central moments of the precipitation frequency distribution should be matched. As this correction is applied to the offline data, it could happen that an already strong precipitation event which might have a scavenging rate of 100% (i.e. all sulphate is already removed by the event) is supposed to remove even more sulphate (which is not available, as it is already depleted!). This is not discussed at all, implying that this precipitation adjustment is a useful measure to correct wet deposition for precipitation

biases.

Overall, I think that this study could be published after addressing the points above, but GMD would have been the better journal.

---

## Author Comment (AC1) · 21 Nov 2019

Response to Comment 1 by Anonymous Reviewer 1

We thank you for providing helpful and constructive comments and suggestions. We have revised our manuscript accordingly. We hope that these revisions satisfactorily address all the points you have raised. Our point-by-point responses are provided below, and revisions are indicated in blue in the revised manuscript.

General Comments:

Authors compare nine meteorology-chemical transport model systems to estimate deposition amount of Sulfur, Nitrogen, and Ammonia. In general, this manuscript is well organized and delivers informative results and an interesting air quality issue over the Northeast Asia. For the model performance evaluations, authors compare the model outputs to the EANET observations. It is reasonable. However, those nine models may already present appreciable differences in airborne concentrations (authors address the companion papers in the text, but short discussion would be helpful for other readers), or may estimate different wet deposition amounts even under the same atmospheric concentrations due to difference in the implemented dry deposition mechanism in the models. Therefore, it would be better to explain more direct relationship between concentrations and dry/wet depositions for model inter-comparisons.

Authors evaluate the precipitation simulations, but wet deposition in CMAQ (6 out of 9 models) not only depends on the precipitation but the types (convective or nonconvective). Moreover, water mixing ratio also plays an important role in CMAQ depending on the meteorological conditions. Those analyses will be very helpful to the air quality research community.

Reply:

We agree that the simulated concentrations are needed to explain the differences in wet deposition identified in this study. However, because this study  is  part of the MICS-Asia project, we prefer not to explicitly show the simulated concentration fields within this manuscript to avoid redundancy with respect to our companion papers that were published in a special issue of MICS-Asia Phase III. Instead, to address your comment, we have added references to our companion papers (Chen et al., 2019; Tan et al., 2019). Additionally, we have evaluated wet deposition at the same observation sites for

consistency and prepared one additional figure (Figure S1 in the revised supporting material) and three additional tables (Tables S1, S2, and S3 in the revised supporting material). These points have now been addressed as independent paragraphs in Sections 3.2.1, 3.2.2, and 3.2.3.

In response to your comment about the precipitation types, please see our reply to your comment about Figure 2.

The revisions for further discussing atmospheric concentrations are as follows:

In Section 3.2.1:

[revised manuscript text omitted]

Scientific Comments:

Table 1: It would be better to include the WRF configurations. Physical options in WRF may affect the wet depositions of those target species.

**Reply:**

**We are grateful for this helpful comment. Because almost all the models considered in this study are driven by the WRF model, we have revised the main manuscript to explicitly mention the WRF configurations. An explanation about the WRF configurations has been added to Section 2.1, as follows:**

**"The WRF is configured as follows: longwave radiation is computed with the rapid radiative transfer model (Mlawer et al., 1997), shortwave radiation with the Goddard scheme (Chou et al., 1994; Matsui et al., 2018), microphysics with Lin's scheme (Chen et al., 2002), cumulus physics with the Grell 3D ensemble scheme (Grell, 1993; Grell and Devenyi, 2002), the planetary boundary layer with the Yonsei University scheme (YSU) (Hong et al., 2006), the surface layer with the revised Fifth-Generation Penn State/NCAR Mesoscale Model (MM5) (Jimenez et al., 2012), and land surface with the unified Noah model (Tewari et al., 2002). The WRF also includes the urban canopy model (Chen et al., 2011)."**

Figure 2: Wet scavenging is affected by not only the precipitation amounts but also precipitation

intensity (and the lasting time), and types (at least in CMAQ parameterization). For example, convective precipitation in MCIP outputs for CMAQ may increase the total rain amounts in summer but may have less influence on fine particle removal, compared to non-convective rain during spring and fall. In case of wet removals of gases species, total surface areas of rain droplets as function of droplet size and ambient bulk concentrations would be important. Size distributions and concentrations of ambient aerosol would be critical for wet scavenging.

**Reply:**

**We agree that precipitation type on the analysis of wet deposition is important. In the framework of the current MICS-Asia Phase III activity, all the submitted results for wet deposition were the sums of wet depositions caused by convective and non-convective precipitation and it is difficult to distinguish between the two types.**

**We will consider this point in the strategy for wet deposition in the next phase of MICS-Asia, which we are now planning. Additionally, we have revised Section 5 to mention this explicitly as follows:**

**"Moreover, precipitation type (convective or non-convective) should be analyzed and the impacts of differences in the characteristics of fine and coarse particles on wet deposition should be investigated."**

Figure 3: Wet depositions in current three dimensional grid models (CMAQ and CAMx) deal with both in-cloud and below-cloud scavenging together. It would be okay to explain total wet depositions, but may mislead to the comparisons to the EANET measurements.

**Reply:**

**EANET ground-based observations use wet-only samplers and measure wet deposition (volume-weighted mean concentrations and precipitation); therefore, these EANET measurement data and model output data as wet deposition could be comparable.**

Figures 3-8: In terms of wet deposition comparisons, Figures and their explanations are well represented in the text. Considering that wet scavenging amounts are determined by airborne concentrations and removal mechanism, it is expected to relate modeled concentrations and wet deposition amounts, including the removal module used in the models.

**Reply:**

**Please see our reply to your 'General Comments'. In the revised manuscript, we have added a detailed discussion of the ambient concentration by appropriately referring to our companion papers, not simply citing them.**

---

## Author Comment (AC2) · 21 Nov 2019

Response to Comment 2 by Anonymous Reviewer 2

Review of MICS-Asia III: Overview of model inter-comparison and evaluation of acid deposition over Asia by Itahashi et al.

**We thank you for providing helpful and constructive comments and suggestions. We have revised our manuscript accordingly. For convenience, we have divided your general comments using numbers. We hope that these revisions have satisfactorily addressed all the points you have raised. Our point-by-point responses are provided below, and revisions are indicated in blue in the revised manuscript.**

1. The paper deals with an intermodel comparison of wet deposition patterns for East Asia given present day conditions. The models used in this study are chemistry transport models and meteorological data is provided by external model fields (for most models form WRF). In general this study is worth publication, however the gain of new scientific knowledge from this effort is rather small. Consequently, would have proposed that this manuscript is much better suited for GMD instead of ACP as a typical evaluation paper. The paper is decently structured and despite some minor required language improvements (proof-reading) could be easily published. The model and observation data description sections are adequate; however if it might be possible to add the observation data file in the supplement more model applicants would benefit from this study.

**Reply:**

**Thank you for your journal recommendation; however, we believe that the ACP is an appropriate selection. The "Aims and Scope" of GMD and ACP are as follows:**

**GMD**

- **geoscientific model descriptions, from statistical models to box models to GCMs;**

- **development and technical papers, describing developments such as new parameterizations or technical aspects of running models such as the reproducibility of results;**

- **new methods for assessment of models, including work developing new metrics for**

assessing model performance and novel ways of comparing model results with observational data;

- papers describing new standard experiments for assessing model performance or novel ways of comparing model results with observational data;

- model experiment descriptions, including experimental details and project protocols;

- full evaluations of previously published models

ACP

ACP covers the altitude range from the land and ocean surface up to the tropopause, including the troposphere, stratosphere, and mesosphere. The main subject areas comprise atmospheric modeling, field measurements, remote sensing, and laboratory studies of gases, aerosols, clouds and precipitation, isotopes, radiation, dynamics, biosphere interactions, and hydrosphere interactions.

The scope of ACP is focused on studies with general implications for atmospheric science, rather than investigations that are primarily of local or technical interest, and authors should thus pay particular attention to whether their paper would fit better in other journals published by Copernicus Publications such as GMD.

We agree that our manuscript may fall within the scope of GMD; however, we would like to submit our manuscript for the special issue of ACP entitled "Regional assessment of air pollution and climate change over East and Southeast Asia: results from MICS-Asia Phase III". Our manuscript describes the Model Inter-comparison Study for Asia (MICS-Asia) Phase III, and the model results are fully evaluated using data from the Acid Deposition Monitoring Network in East Asia (EANET) covering the whole of Asia. The manuscript contains not only model evaluation, but also the following three main discussion points: 1. acid deposition mapping over Asia; 2. an ensemble approach; and 3. a precipitation-adjusted approach. The latter two discussion points are aimed at improving modeling performance, whereas the first point is strongly related to the importance of the EANET observation network, as well as the balance between emissions (input) and depositions (output). Our manuscript concerns the impacts of Asian air

quality not only locally but also globally, so we believe that it has general implications for atmospheric science rather than investigations that are primarily of local or technical interest and that it will be of interest to readers of ACP. To increase the suitability of our manuscript for publication in ACP, we have sought to further improve it based on the reviewers' comments.

The EANET observational dataset used in this study is freely available from a publicly accessible website. Therefore, we believe that it is not necessary to include data as supplemental material. However, we have added a "Data Availability" section to the end of the revised manuscript in which we provide the URL where the EANET dataset can be accessed.

2. Despite the model domain is very similar (if not identical) in the simulations and the models use unified emissions, substantial differences in the deposition maps occur. Unfortunately, the authors do not show a comparison of the simulated concentrations or tropospheric vertical columns of the species the study focuses on. Therefore, it is difficult to judge, whether the differences occur from slightly different assumptions in the wet deposition schemes, the simulated precipitation amount or the simulated concentrations of the trace species and their precursors. As the applied chemistry and aerosol schemes differ to a certain degree, this could already be a major cause for the differences in the wet deposition patterns.

**Reply:**

**We agree that the simulated concentrations are needed to explain the differences in wet deposition analyzed in this study. Because our result is a part of the MICS-Asia project, we prefer not to explicitly show the simulated concentration fields within this manuscript to avoid redundancy with respect to our companion papers that were published in a special issue of MICS-Asia Phase III. Instead, to address your comment, we have added references to our companion papers (Chen et al., 2019; Tan et al., 2019). Additionally, we have conducted an evaluation of wet deposition to be consistent in the use of the same observation sites and prepared one additional figure (Figure S1 in the revised supporting material) and three additional tables (Tables S1, S2, and S3 in the revised supporting material). These points have now been addressed as independent**

paragraphs in Sections 3.2.1, 3.2.2, and 3.2.3.

In Section 3.2.1:

[revised manuscript text omitted]

3. The evaluation of the precipitation is unfortunately only superficial. As only monthly mean precipitation is compared with the simulation results, the corresponding frequency distribution, i.e. the number, duration and intensity of the events cannot be determined. However, this is crucial for wet deposition, as a few short but intense rain events result in less deposition compared to longer precipitation events of average intensity. This will also substantially impact the precipitation adjustment (see comment below!).

**Reply:**

**We agree that the evaluation of monthly mean precipitation is not appropriate in terms of the frequency distribution, number, duration, and intensity of rain events. However, the sampling interval of wet deposition is mostly daily over North and East Asia and weekly over Southeast Asia (Table 2). Additionally, the sampling periods of these measurements were not consistent across site. To allow for a consistent analysis period for the whole dataset, we believe that taking the monthly-mean precipitation is an appropriate approach and this analysis could provide a broad overview in Asia.**

**In response to your comment about the impact on precipitation adjustment, please see our reply to comment 8.**

4. Analyzing the wet deposition of sulphur, M11 shows a substantially higher deposition pattern in China. What is the reason for this? This is a typical example for a model inter-comparison study where data is compared, but the causes for the differences are not analysed in detail. Has there been an issue with SO2 emissions or conversion from S(IV) to S(VI)? Is there a bias originating from seasalt sulphate? Is total Sulphur completely overestimated in this model? Or is it completely depleted, as wet deposition is so efficient? These differences require much more analysis for a consistent intercomparison study.

**Reply:**

**This is related to our reply to comment 2. We have added a statement about the sulfur**

**oxidation ratio by referring to our companion paper (Tan et al., 2019).**

5. Especially, when creating ensembles including such outliers, the ensemble mean can even be deteriorated compared to individual simulations. This does not appear to be the case in this study, as the M11 simulation compensates some of the low bias from the majority of the other simulations. A similar behaviour of overestimation is not as obvious for nitrate and ammonium, leading to the impression that this is not necessarily a consequence of the wet deposition scheme.

> **Reply:**
>
> **We think that the overestimation of the wet deposition of N is also noticeable in model M11. For the wet deposition of S, model M11 was the only model that overestimated (all of the other models underestimated); however, in the case of the wet deposition of N, there was greater variation in the performance of all models. As a result, the ensemble mean for the wet deposition of N performed better than that of individual models.**

6. The weighted ensemble might be a better option to reduce the importance of outliers; however, it simply states that the models which show best agreement with the observations should be used for the ensemble mean. Consequently, it reduces ensemble spread and therefore does not cover the whole range of simulation results properly. Please state explicitly, what you hope to gain from the weighted ensemble mean.

> **Reply:**
>
> **In this weighted ensemble mean, we used R as the weighting factor. Therefore, we did not entirely eliminate the model that showed lower performance than observations.**
>
> **Near the end of Section 4.2, we clearly state the superiority of the weighted ensemble mean as follows:**
>
> **"In terms of NMB, ENS performed better than WENS; however, WENS could be regarded as a better approach because it takes into account each model performance evaluated by observation using R as the weighting factor and it showed better values than ENS in terms of NME, FAC2, and FAC3."**

7. Concerning the total deposition maps, the authors should clearly point out, that underestimated wet deposition can often be compensated by overestimated dry deposition and vice versa, as both processes depend on the atmospheric burden (or near surface concentrations).

**Reply:**

**We appreciate this constructive comment. We have added the following sentence in Section 4.1, accordingly:**

**"As we have seen (e.g., Fig. S1), the underestimation (overestimation) of wet deposition could be related to the overestimation (underestimation) of atmospheric concentration, and these could be found as dry deposition. The underestimation (overestimation) of wet deposition can be compensated by the overestimation (underestimation) of dry deposition, and may pose the similar total deposition amount. Therefore, this kind of study can give insights into the balance between dry and wet deposition."**

8. I (personally) see the option of precipitation adjustment to improve the consistency of the simulation results with observations very critical. This adjustment does not include any kind of frequency distribution of precipitation events, the vertical extent of the precipitation (and hence the accessible fraction of the tracer vertical column for wet deposition). Also it does not include any kind of vertical redistribution by scavenging and subsequent evaporating precipitation and hence tracer release at lower altitude. Of course, I agree that with wrong precipitation amounts it will be impossible to fully match observations, but in my opinion not only the total amount of precipitation, but at least the central moments of the precipitation frequency distribution should be matched. As this correction is applied to the offline data, it could happen that an already strong precipitation event which might have a scavenging rate of 100% (i.e. all sulphate is already removed by the event) is supposed to remove even more sulphate (which is not available, as it is already depleted!). This is not discussed at all, implying that this precipitation adjustment is a useful measure to correct wet deposition for precipitation biases.

**Reply:**

**We agree with your comments on the adjustment approach used in this study. To avoid overstating the usefulness of this approach, we have revised the introduction to this method in Section 4.3 as follows:**

**"Note that this precipitation-adjusted approach assumes that errors associated with the modeled precipitation are linearly related to errors in wet deposition amounts and the precipitation was adjusted by the total amount of observed precipitation; hence, the modeled convective and sub-grid-scale precipitation was not distinguished. Because current meteorological models have difficulty in capturing the timing of precipitation events, the application of this adjusted approach at a finer temporal resolution will lead to excessive adjustments (e.g., close to zero in the case that observed precipitation is zero or divergent in the case that the modeled precipitation is near zero)."**

9. Overall, I think that this study could be published after addressing the points above, but GMD would have been the better journal.

> **Reply:**
>
> **We have addressed the appropriateness of our submission for ACP rather than GMD in our reply to comment 1.**